# RETRO-FALLBACK: RETROSYNTHETIC PLANNING IN AN UNCERTAIN WORLD

**Austin Tripp**[1]* **Krzysztof Maziarz**[2] **Sarah Lewis**[2]
**Marwin Segler**[2] **José Miguel Hernández-Lobato**[1]
[1]University of Cambridge    [2]Microsoft Research AI4Science
{ajt212,jmh233}@cam.ac.uk
{krmaziar,sarahlewis,marwinsegler}@microsoft.com

## ABSTRACT

Retrosynthesis is the task of planning a series of chemical reactions to create a desired molecule from simpler, buyable molecules. While previous works have proposed algorithms to find optimal solutions for a range of metrics (e.g. shortest, lowest-cost), these works generally overlook the fact that we have imperfect knowledge of the space of possible reactions, meaning plans created by algorithms may not work in a laboratory. In this paper we propose a novel formulation of retrosynthesis in terms of stochastic processes to account for this uncertainty. We then propose a novel greedy algorithm called retro-fallback which maximizes the probability that at least one synthesis plan can be executed in the lab. Using *in-silico* benchmarks we demonstrate that retro-fallback generally produces better sets of synthesis plans than the popular MCTS and retro* algorithms.

## 1 INTRODUCTION

Retrosynthesis (planning the synthesis of organic molecules via a series of chemical reactions) is a common task in chemistry with a long history of automation (Vleduts, 1963; Corey & Wipke, 1969). Although the combinatorially large search space of chemical reactions makes naive brute-force methods ineffective, recently significant progress has been made by developing modern machine-learning based search algorithms for retrosynthesis (Strieth-Kalthoff et al., 2020; Tu et al., 2023; Stanley & Segler, 2023). However, there remain obstacles to translating the output of retrosynthesis algorithms into real-world syntheses. One significant issue is that these algorithms have imperfect knowledge of the space of chemical reactions. Because the underlying physics of chemical reactions cannot be efficiently simulated, retrosynthesis algorithms typically rely on data-driven reaction prediction models which can "hallucinate" unrealistic or otherwise infeasible reactions (Zhong et al., 2023). This results in synthesis plans which cannot actually be executed.

Although future advances in modelling may reduce the prevalence of infeasible reactions, we think it is unlikely that they will ever be eliminated entirely, as even the plans of expert chemists do not always work on the first try. One possible workaround to failing plans is to produce *multiple* synthesis plans instead of just a single one: the other plans can act as *backup* plans in case the primary plan fails. Although existing algorithms may find multiple synthesis plans, they are generally not designed to do so, and there is no reason to expect the plans found will be suitable as *backup* plans (e.g. they may share steps with the primary plan and thereby also share the same failure points).

In this paper, we present several advancements towards retrosynthesis with backup plans. First, in section 3 we explain how uncertainty about whether a synthesis plan will work in the lab can be quantified with stochastic processes. We then propose an evaluation metric called *successful synthesis probability* (SSP) which quantifies the probability that *at least one* synthesis plan found by an algorithm will work. This naturally captures the idea of producing backup plans. Next, in section 4 we present a novel search algorithm called *retro-fallback* which greedily optimizes

---

*Work done partly during internship at Microsoft Research AI4Science

SSP. Finally, in section 6 we demonstrate quantitatively that retro-fallback outperforms existing algorithms on several *in-silico* benchmarks. Together, we believe these contributions form a notable advancement towards translating results from retrosynthesis algorithms into the lab.

## 2 BACKGROUND: STANDARD FORMULATION OF RETROSYNTHESIS

Let $\mathcal{M}$ denote the space of molecules, and $\mathcal{R}$ denote the space of single-product reactions which transform a set of *reactant* molecules in $2^{\mathcal{M}}$ into a *product* molecule in $\mathcal{M}$. The set of reactions which produce a given molecule is given by a *backward reaction model* $B : \mathcal{M} \mapsto 2^{\mathcal{R}}$. $B$ can be used to define an (implicit) reaction graph $\mathcal{G}$ with nodes for each molecule and each reaction, and edges linking molecules to reactions which involve them. Figure 1a illustrates a small example graph. Note that by convention the arrows are drawn backwards (from products towards reactants). This kind of graph is sometimes called an *AND/OR graph* (see Appendix B for details).

A *synthesis plan* for a molecule $m$ is a sequence of chemical reactions which produces $m$ as the final product. Synthesis plans usually form trees $T \subseteq \mathcal{G}$ (more generally directed acyclic subgraphs), wherein each molecule is produced by at most one reaction. The set of all synthesis plans in $\mathcal{G}$ which produce a molecule $m$ is denoted $\mathcal{P}_m(\mathcal{G})$. Figure 1b provides an example (see Appendix B.2 for a detailed definition). Not all synthesis plans are equally useful however. Most importantly, for a synthesis plan to actually be executed by a chemist the starting molecules must all be bought. Typically this is formalized as requiring all starting molecules to be contained in an *inventory* $\mathcal{I} \subseteq \mathcal{M}$ (although we will propose an alternative formulation in section 3). It is also desirable for synthesis plans to have low cost, fewer steps, and reactions which are easier to perform.

In retrosynthesis, one usually seeks to create synthesis plans for a specific *target molecule* $m_\star$. Typically this is formulated as a search problem over $\mathcal{G}$. Various search algorithms have been proposed which, at a high level, all behave similarly. First, they initialize an *explicit subgraph* $\mathcal{G}' \subseteq \mathcal{G}$ with $\mathcal{G}' \leftarrow \{m_\star\}$. Nodes whose children have not been added to $\mathcal{G}'$ form the *frontier* $\mathcal{F}(\mathcal{G}')$. Then, at each iteration $i$ they select a frontier molecule $m_{(i)} \in \mathcal{F}(\mathcal{G}')$ (necessarily $m_\star$ on the first iteration), query $B$ to find reactions which produce $m_{(i)}$, then add these reactions and their corresponding reactant molecules to the explicit graph $\mathcal{G}'$. This process is called *expansion*, and is illustrated for $m_c$ in Figure 1a. Search continues until a suitable synthesis plan is found or until the computational budget is exhausted. Afterwards, synthesis plans can be enumerated from $\mathcal{G}'$.

The most popular retrosynthesis search algorithms compute some sort of metric of synthesis plan quality, and use a *search heuristic* to guide the search towards high-quality synthesis plans. For example, Monte Carlo Tree Search (*MCTS*) searches for synthesis plans which maximize an arbitrary scalar reward function (Segler et al., 2018). *Retro\** is a best-first search algorithm to find minimum-cost synthesis plans, where the cost of a synthesis plan is defined as the sum of costs for each reaction and each starting molecule (Chen et al., 2020). In both algorithms, frontier nodes are chosen using the heuristic to estimate the reward (or cost) which could be achieved upon expansion. We introduce these algorithms more extensively in Appendix E.

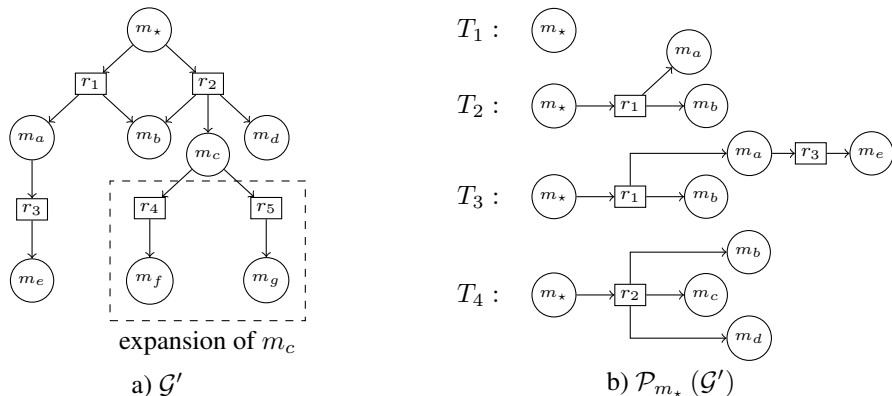

Figure 1: **a)** graph $\mathcal{G}'$ with (backward) reactions $m_\star \Rightarrow m_a + m_b$ $(r_1)$, $m_\star \Rightarrow m_b + m_c + m_d$ $(r_2)$, and $m_a \Rightarrow m_e$ $(r_3)$. Dashed box illustrates expansion of $m_c$. **b)** All synthesis plans in $\mathcal{P}_{m_\star}(\mathcal{G}')$.

## 3 REFORMULATING RETROSYNTHESIS WITH UNCERTAINTY

The "standard" formulation of retrosynthesis presented in section 2 requires knowledge of which reactions are possible (encoded by the backward reaction model $B$) and which molecules are purchasable (encoded by the inventory $\mathcal{I}$). In reality, neither of these things are perfectly known. As mentioned in the introduction, predicting the outcome of chemical reactions is difficult even for experts, and machine learning models for $B$ can "hallucinate" unrealistic reactions. Perhaps surprisingly, it is also not totally clear which molecules can be bought. Things like shipping delays mean you might not always receive molecules which you order. However, many companies now advertise large "virtual libraries" with billions of molecules which they *believe* they can synthesize upon request, but not with 100% reliability.[1] This section presents our first main contribution to account for this: a novel formulation of retrosynthesis which explicitly represents uncertainty.

### 3.1 STOCHASTIC PROCESSES FOR "FEASIBILITY" AND "BUYABILITY"

There are many reasons why chemists may consider a reaction unsuccessful, ranging from having a low yield to producing the wrong product altogether. Similarly, "unsuccessfully" buying a molecule could indicate anything from a prohibitively high cost to the molecule not being delivered. In either case, for simplicity we propose to collapse this nuance into a binary outcome: reactions are either *feasible* or *infeasible*, and molecules are either *buyable* or not. We therefore postulate the existence of an unknown "feasibility" function $f^* : \mathcal{R} \mapsto \{0, 1\}$ and "buyability" function $b^* : \mathcal{M} \mapsto \{0, 1\}$.

Uncertainty about $f^*$ and $b^*$ can be represented by *stochastic processes* (essentially distributions over functions). We define a *feasibility model* $\xi_f$ to be a binary stochastic process over $\mathcal{R}$, and define a *buyability model* $\xi_b$ to be a binary stochastic process over $\mathcal{M}$. This formulation is very general: $\xi_f$ and $\xi_b$ not only represent beliefs of $\mathbb{P}[f^*(r) = 1]$ and $\mathbb{P}[b^*(m) = 1]$ for all molecules $m$ and reactions $r$, but also allows *correlations* between feasibilities and buyabilities to be modelled.

Although this formalism may seem esoteric, it is possible to re-cast almost all existing approaches to reaction prediction as stochastic processes. Any model which implicitly assigns a probability to each reaction (e.g. the softmax outputs of a neural network) can be trivially converted into a stochastic process by assuming that all outcomes are independent. Correlations can be induced via Bayesian inference over the model's parameters (MacKay, 1992) or using a non-parametric model like a Gaussian process (Williams & Rasmussen, 2006). Importantly however, it is not at all clear how to produce *realistic* models $\xi_f$ and $\xi_b$. Intuitively, producing such models is at least as challenging as predicting reaction outcomes *without* uncertainty estimates, which is itself an active (and challenging) research area. Therefore, we will generally discuss $\xi_f / \xi_b$ in a model-agnostic way.

### 3.2 NEW EVALUATION METRIC: SUCCESSFUL SYNTHESIS PROBABILITY (SSP)

Given $f$ and $b$, a synthesis plan $T$ is *successful* if all its reactions $r$ are feasible ($f(r) = 1$) and all its starting molecules $m$ are buyable ($b(m) = 1$). We formalize this with the function

$$\sigma(T; f, b) = \begin{cases} 1 & f(r) = 1 \; \forall r \in T, \; b(m) = 1 \text{ and } \forall m \in \mathcal{F}(T) \\ 0 & \text{otherwise} \end{cases}. \tag{1}$$

Finding successful synthesis plans is a natural goal of retrosynthesis. Of course, because $f$ and $b$ are unknown, we can at best search for synthesis plans with a high *probability* of being successful. Given a *set* of synthesis plans $\mathcal{T}$, we define the *successful synthesis probability* (SSP) as:

$$\text{SSP}(\mathcal{T}; \xi_f, \xi_b) = \mathbb{P}_{f \sim \xi_f, b \sim \xi_b} [\exists \, T \in \mathcal{T} \text{ with } \sigma(T; f, b) = 1] \tag{2}$$

Given just a single plan $T$, $\text{SSP}(\{T\}; \xi_f, \xi_b) = \mathbb{E}_{f,b} [\sigma(T; f, b)]$ and represents the probability that $T$ is successful, which we will hereafter refer to as the *success probability* of $T$. When $\mathcal{T}$ contains multiple synthesis plans, then SSP quantifies the probability that *any* of these synthesis plans is successful. We argue that SSP is a good evaluation metric for the synthesis plans produced by retrosynthesis search algorithms. It simultaneously captures the goals of producing synthesis plans with high success probability and producing "backup" plans which could succeed if the primary synthesis plan does not. Note that by definition, SSP is non-decreasing with respect to $\mathcal{T}$, implying that an algorithm will never be penalized for producing additional synthesis plans.

---

[1]For example, Enamine a popular supplier, only claims that 80% of its virtual "REAL" library can be made.

### 3.3 Efficiently estimating SSP for all synthesis plans in $\mathcal{P}_{m_\star}(\mathcal{G}')$

Recall from section 2 that many retrosynthesis search algorithms do not directly output synthesis plans: they produce a search graph $\mathcal{G}'$ which (implicitly) contains a set of synthesis plans $\mathcal{P}_{m_\star}(\mathcal{G}')$. Therefore, it is natural to calculate the SSP of the entire set $\mathcal{P}_{m_\star}(\mathcal{G}')$. However, this set may be combinatorially large, making calculating SSP by enumerating $\mathcal{P}_{m_\star}(\mathcal{G}')$ intractable. Instead, we propose a method to estimate SSP using functions sampled from $\xi_f$ and $\xi_b$.

Let $s(n; \mathcal{G}', f, b) : \mathcal{M} \cup \mathcal{R} \mapsto \{0, 1\}$ define the *success* of a node $n \in \mathcal{G}$: whether *any* successful synthesis plan in $\mathcal{G}$ contains $n$ (we write $s(n)$ when $\mathcal{G}', f, b$ are clear from context). $s(n)$ will satisfy

$$s(n; \mathcal{G}', f, b) \overset{(A)}{=} \sigma(T^*; f, b) \overset{(B)}{=} s(n; T^*, f, b), \qquad T^* \in \underset{T \in \mathcal{P}_*(\mathcal{G}'): \, n \in T}{\arg \max} \sigma(T; f, b), \qquad (3)$$

where $\mathcal{P}_*(\mathcal{G}') = \bigcup_{m \in \mathcal{G}'} \mathcal{P}_m(\mathcal{G}')$ is the set of all synthesis plans for all molecules in $\mathcal{G}'$. Equality (A) follows directly from the definition above, and equality (B) holds because $T^*$ would still satisfy the $\arg \max$ if nodes not in $T^*$ were pruned from $\mathcal{G}'$. Let $Ch_{\mathcal{G}'}(n)$ denote the children of node $n$. For a reaction $r \in \mathcal{G}'$ to succeed, it must be feasible ($f(r) = 1$) and have all its reactant molecules $m' \in Ch_{\mathcal{G}'}(r)$ succeed. Conversely, a molecule $m \in \mathcal{G}'$ will succeed if it is buyable ($b(m) = 1$) or if any reaction producing $m$ succeeds. This suggests $s(\cdot)$ will satisfy the recursive equations

$$s(m; \mathcal{G}', f, b) = \max \left[ b(m), \max_{r \in Ch_{\mathcal{G}'}(m)} s(r; \mathcal{G}', f, b) \right], \qquad (4)$$

$$s(r; \mathcal{G}', f, b) = f(r) \prod_{m \in Ch_{\mathcal{G}'}(r)} s(m; \mathcal{G}', f, b). \qquad (5)$$

SSP can then be estimated by averaging $s(m_\star)$ over $k$ i.i.d. functions sampled from $\xi_f$ and $\xi_b$:

$$\mathrm{SSP}(\mathcal{P}_{m_\star}(\mathcal{G}'); \xi_f, \xi_b) \overset{(A)}{=} \mathbb{P}_{f \sim \xi_f, b \sim \xi_b} \left[ s(m_\star; \mathcal{G}', f, b) = 1 \right] \approx \frac{1}{k} \sum_{i=1}^{k} s(m_\star; \mathcal{G}', f_k, b_k). \qquad (6)$$

Note that equality (A) above follows directly from equations 2 and 3. The existence of such recursive equations suggests that $s(\cdot)$ could be efficiently computed for all nodes in $\mathcal{G}'$ in polynomial time using dynamic programming (we discuss this further in Appendix D.2), allowing an overall polynomial time estimate of SSP. That being said, it is still only an *estimate*. Unfortunately, we are able to prove that an exact calculation is generally intractable.

**Theorem 3.1.** *Unless $P = NP$, there does not exist an algorithm to compute $\mathrm{SSP}(\mathcal{P}_{m_\star}(\mathcal{G}'); \xi_f, \xi_b)$ for arbitrary $\xi_f, \xi_b$ whose time complexity grows polynomially with the number of nodes in $\mathcal{G}'$.*

The proof is given in Appendix D.1. We therefore conclude that estimating SSP using equation 6 is the best realistic option given limited computational resources.

## 4 Retro-fallback: a greedy algorithm to maximize SSP

### 4.1 Ingredients for an informed, greedy search algorithm

Intuitively, a greedy search algorithm would expand molecules in $\mathcal{F}(\mathcal{G}')$ which are predicted to improve SSP. Given that calculating SSP exactly is intractable, calculating potential changes is likely to be intractable as well. Therefore, we will estimate SSP changes by averaging over samples from $\xi_f$ and $\xi_b$, and will consider how expansion might change $s(m_\star; \mathcal{G}', f, b)$ for fixed samples $f, b$.

Specifically, we consider the effect of simultaneously expanding *every* frontier molecule on a fixed synthesis plan $T \in \mathcal{P}_*(\mathcal{G}')$.[2] We represent the hypothetical effect of such an expansion with a random function $e_T : \mathcal{M} \mapsto \{0, 1\}$, where $e_T(m) = 1$ implies that expanding $m$ produces a new successful synthesis plan for $m$. We assume the value of $e_T$ is independently distributed for every molecule, with probabilities given by a *search heuristic* function $h : \mathcal{M} \mapsto [0, 1]$

$$\mathbb{P}_{e_T} \left[ e_T(m) = 1 \right] = \begin{cases} h(m) & m \in \mathcal{F}(\mathcal{G}') \cap T \\ 0 & m \notin \mathcal{F}(\mathcal{G}') \cap T \end{cases}. \qquad (7)$$

---

[2]We do not consider expanding just a single node because, for a reaction with multiple non-buyable reactant molecules in $\mathcal{F}(\mathcal{G}')$, expanding just *one* reactant will never produce a new successful synthesis plan.

The effect of this expansion on the success of $T$ is given by $\sigma' : \mathcal{P}_*(\mathcal{G}') \mapsto \{0, 1\}$, defined as

$$\sigma'(T; f, b, e_T) = \begin{cases} 1 & f(r) = 1 \ \forall r \in T \text{ and } (b(m) = 1 \text{ or } e_T(m) = 1) \ \forall m \in \mathcal{F}(T) \\ 0 & \text{otherwise} \end{cases} . \quad (8)$$

Equation 8 for $\sigma'$ is almost identical to equation 1 for $\sigma$. The key difference (highlighted) is that $T$ can be successful if a starting molecule $m$ is not buyable ($b(m) = 0$) but has instead had $e_T(m) = 1$. Recalling that $e_T$ is a random function, we define $\bar{\sigma}' : \mathcal{P}_*(\mathcal{G}') \mapsto [0, 1]$ as

$$\bar{\sigma}'(T; f, b, h) = \mathbb{E}_{e_T} \left[ \sigma'(T; f, b, e_T) \right] , \quad (9)$$

namely the *probability* that a synthesis plan $T$ will be successful upon expansion.[3] A natural choice for a greedy algorithm could be to expand frontier nodes on synthesis plans $T$ with high $\bar{\sigma}'(T; f, b, h)$. However, not all synthesis plans contain frontier nodes (e.g. plan $T_1$ in Figure 1b) or produce $m_\star$. To select frontier nodes for expansion, we define the function $\tilde{\rho} : \mathcal{M} \cup \mathcal{R} \mapsto [0, 1]$ by

$$\tilde{\rho}(n; \mathcal{G}', f, b, h) = \max_{T \in \mathcal{P}_{m_\star}(\mathcal{G}'): \ n \in T} \bar{\sigma}'(T; f, b, h) , \qquad n \in \mathcal{G}' . \quad (10)$$

For $m \in \mathcal{F}(\mathcal{G}')$, $\tilde{\rho}(m)$ represents the highest estimated success probability of all synthesis plans for $m_\star$ which also contain $m$ (conditioned on a particular $f, b$). Therefore, a greedy algorithm could sensibly expand frontier molecules $m$ with maximal $\tilde{\rho}(m)$.

Unfortunately, the combinatorially large number of synthesis plans in a graph $\mathcal{G}'$ makes evaluating $\tilde{\rho}$ potentially infeasible. To circumvent this, we assume that no synthesis plan in $\mathcal{G}'$ uses the same molecule in two separate reactions, making all synthesis plans trees (we will revisit this assumption later). This assumption guarantees that the outcomes from different branches of a synthesis plan will always be independent. Then, to help efficiently compute $\tilde{\rho}$, we will define the function

$$\tilde{\psi}(n; \mathcal{G}', f, b, h) = \max_{T \in \mathcal{P}_*(\mathcal{G}'): \ n \in T} \bar{\sigma}'(T; f, b, h) \quad (11)$$

for every node $n \in \mathcal{G}'$. $\tilde{\psi}$ is essentially a less constrained version of $\tilde{\rho}$. The key difference in their definitions is that $\tilde{\psi}$ maximizes over *all* synthesis plans containing $n$, including plans which do not produce $m_\star$. The independence assumption above means that $\tilde{\psi}$ has a recursively-defined analytic solution $\psi(\cdot; \mathcal{G}', f, b, h) : \mathcal{M} \cup \mathcal{R} \mapsto [0, 1]$ given by the equations

$$\psi(m; \mathcal{G}', f, b, h) = \begin{cases} \max\left[b(m), h(m)\right] & m \in \mathcal{F}(\mathcal{G}') \\ \max\left[b(m), \max_{r \in Ch_{\mathcal{G}'}(m)} \psi(r; \mathcal{G}', f, b, h)\right] & m \notin \mathcal{F}(\mathcal{G}') \end{cases} , \quad (12)$$

$$\psi(r; \mathcal{G}', f, b, h) = f(r) \prod_{m \in Ch_{\mathcal{G}'}(r)} \psi(m; \mathcal{G}', f, b, h) . \quad (13)$$

Details of this solution are presented in Appendix C.1. $\psi(n)$ can be roughly interpreted as "the best expected success value for $n$ upon expansion." In fact, the relationship between $\psi$ and $\bar{\sigma}'$ is exactly analogous to the relationship between s and $\sigma$ in equation 3.

To compute $\tilde{\rho}$, first note that $\tilde{\rho}(m_\star) = \tilde{\psi}(m_\star)$, as for $m_\star$ the constraints in equations 10 and 11 are equivalent. Second, because of the independence assumption above, the best synthesis plan containing *both* a node $n$ and its parent $n'$ can be created by taking an optimal synthesis plan for $n'$ (which may or may not contain $n$), removing the part "below" $n'$, and adding in an (unconstrained) optimal plan for $n$. Letting $Pa_{\mathcal{G}'}(\cdot)$ denote a node's parents,[4] under this assumption $\tilde{\rho}$ has a recursively-defined analytic solution $\rho(\cdot; \mathcal{G}', f, b, h) : \mathcal{M} \cup \mathcal{R} \mapsto [0, 1]$ defined as

$$\rho(m; \mathcal{G}', f, b, h) = \begin{cases} \psi(m; \mathcal{G}', f, b, h) & m \text{ is target molecule } m_\star \\ \max_{r \in Pa_{\mathcal{G}'}(m)} \rho(r; \mathcal{G}', f, b, h) & \text{all other } m \end{cases} , \quad (14)$$

$$\rho(r; \mathcal{G}', f, b, h) = \begin{cases} 0 & \psi(r; \mathcal{G}', f, b, h) = 0 \\ \rho(m'; \mathcal{G}', f, b, h) \frac{\psi(r; \mathcal{G}', f, b, h)}{\psi(m'; \mathcal{G}', f, b, h)} & \psi(r; \mathcal{G}', f, b, h) > 0, m' \in Pa_{\mathcal{G}'}(r) \end{cases} . \quad (15)$$

---

[3]The dependence on $h$ is because it defines the distribution of $e_T$ in equation 8.

[4]Recall that because we consider only single-product reactions, all reaction nodes will have exactly one parent, making equation 15 well-defined.

Details of this solution are presented in Appendix C.1. Like $s(\cdot)$, $\psi$ and $\rho$ have recursive definitions, and can therefore be calculated with dynamic programming techniques. Since $\psi$ depends on a node's children, it can generally be calculated "bottom-up", while $\rho$ can be calculated "top-down" because it depends on a node's parents. We discuss details of computing $\psi$ and $\rho$ in Appendix C.1, and provide a full worked-through example in Appendix C.2.

However, in deriving $\psi$ and $\rho$ we assumed that all synthesis plans $T \in \mathcal{P}_*(\mathcal{G}')$ were trees. In practice, this assumption may not hold (see Figure C.1 for an example). If this assumption is violated, $\psi$ and $\rho$ can both still be calculated, but will effectively *double-count* molecules which occur multiple times in a synthesis plan, and therefore not equal $\tilde{\psi}$ and $\tilde{\rho}$. This is a well-known issue in AND/OR graphs: for example, Nilsson (1982, page 102) describes the essentially same issue when calculating minimum cost synthesis plans. Ultimately we will simply accept this and use $\psi/\rho$ instead of $\tilde{\psi}/\tilde{\rho}$ despite their less principled interpretation, chiefly because the recursive definitions of $\psi/\rho$ are amenable to efficient computation. Synthesis plans which use the same molecule twice are unusual in chemistry; therefore we do not expect this substitution to be problematic in practice.

## 4.2 RETRO-FALLBACK: A FULL GREEDY ALGORITHM

Recall our original goal at the start of section 4.1: to estimate how expansion might affect SSP. We considered *a single sample* $f \sim \xi_f$ and $b \sim \xi_b$, and developed the function $\rho$, which for each frontier molecule $m \in \mathcal{F}(\mathcal{G}')$ gives the best estimated synthesis plan for $m_\star$ if $m$ is expanded (simultaneously along with other frontier molecules on an optimally chosen synthesis plan). We will now use $\rho$ to construct a full algorithm.

Expanding a frontier molecule can improve SSP if, for samples $f$ and $b$ where $s(m_\star; \mathcal{G}', f, b) = 0$, the expansion changes this to 1. In this scenario, expanding a frontier molecule $m^* \in \arg\max_{m \in \mathcal{F}(\mathcal{G}')} \rho(m; \mathcal{G}', f, b, h)$ is a prudent choice, as it lies on a synthesis plan with the highest probability of "flipping" $s(m_\star; \mathcal{G}', f, b)$ to 1. In contrast, because $s(\cdot)$ will never decrease as nodes are added, if $s(m_\star; \mathcal{G}', f, b) = 1$ then it does not matter which molecule is expanded. Therefore, when aggregating over samples of $f$ and $b$ to decide which molecules to expand to improve SSP, we will consider the value of $\rho$ *only* in cases when $s(m_\star; \mathcal{G}', f, b) = 0$.

For our greedy algorithm, we propose to simply expand the molecule with the highest *expected improvement* of SSP. Letting $\mathbf{1}_{(\cdot)}$ be the indicator function, this is a molecule $m \in \mathcal{F}(\mathcal{G}')$ which maximizes

$$\alpha(m; \mathcal{G}', \xi_f, \xi_b, h) = \mathbb{E}_{f \sim \xi_f, b \sim \xi_b} \left[ \mathbf{1}_{s(m_\star; \mathcal{G}', f, b)=0} \left[ \rho(m; \mathcal{G}', f, b, h) \right] \right] \tag{16}$$

In practice, $\alpha$ would be estimated from a finite number of samples from $\xi_f$ and $\xi_b$. Using $\rho$ to select a *single* molecule may seem odd, especially because $\rho$ is defined as a hypothetical outcome of simultaneously expanding multiple nodes. However, note that in principle there is nothing problematic about expanding these nodes one at a time.

We call our entire algorithm *retro-fallback* (from "retrosynthesis with fallback plans") and state it explicitly in Algorithm 1. The sections are colour-coded for clarity. After initializing $\mathcal{G}'$, the algorithm performs $L$ iterations of expansion (although this termination condition could be changed as needed). In each iteration, first the values of s, $\psi$, and $\rho$ are computed for each sample of $f$ and $b$.[5] Next, the algorithm checks whether there are no frontier nodes or whether the estimated SSP is 100%, and if so terminates (both of these conditions mean no further improvement is possible). Finally, a frontier node maximizing $\alpha$ (16) is selected and expanded. Of course, a practical implementation of retro-fallback may look slightly different from Algorithm 1. We refer the reader to Appendix C for further discussion about the design and implementation of retro-fallback.

## 5 RELATED WORK

Retro-fallback is most comparable with other retrosynthesis search algorithms including MCTS (Segler et al., 2018), retro* (Chen et al., 2020), and proof number search (Heifets & Jurisica, 2012; Kishimoto et al., 2019). At a high level these algorithms are all similar: they use a heuristic to

---

[5]This order is chosen because s depends only on $f$ & $b$, $\psi$ depends on s, and $\rho$ depends on $\psi$. Because the optimal algorithm to compute s, $\psi$, $\rho$ may depend on $\mathcal{G}'$, we only specify this computation generically.

---

**Algorithm 1** Retro-fallback algorithm (see 4.2)

---

**Require:** target molecule $m_\star$, max iterations $L$, backward reaction model $B$, search heuristic $h$
**Require:** samples $f_1, \ldots, f_k \sim \xi_f$, $b_1, \ldots, b_k \sim \xi_b$
1: $\mathcal{G}' \leftarrow \{m_\star\}$
2: **for** $i$ in $1, \ldots, L$ **do**
3:     **for** $j$ in $1, \ldots, k$ **do**
4:         Compute $s(\cdot; \mathcal{G}', f_j, b_j)$ for all nodes using equations 4–5
5:         Compute $\psi(\cdot; \mathcal{G}', f_j, b_j, h)$ for all nodes using equations 12–13
6:         Compute $\rho(\cdot; \mathcal{G}', f_j, b_j, h)$ for all nodes using equations 14–15
7:     **end for**
8:     Terminate early if $|\mathcal{F}(\mathcal{G}')| = 0$ OR $s(m_\star; \mathcal{G}', f_j, b_j) = 1 \, \forall j$
9:     $m_{(i)} \leftarrow \arg\max_{m \in \mathcal{F}(\mathcal{G}')} \alpha(m; \mathcal{G}', \xi_f, \xi_b, h)$ (equation 16, breaking ties arbitrarily)
10:     Add all reactions and molecules from $B(m_{(i)})$ to $\mathcal{G}'$
11: **end for**
12: **return** $\mathcal{G}'$

---

guide the construction of an explicit search graph. However, previous algorithms may struggle to maximize SSP because their internal objectives consider only *individual* synthesis plans, while SSP depends on *multiple* synthesis plans simultaneously. In Appendix E.2 we argue that for most algorithms the best proxy for SSP is the success probability of individual synthesis plans, but illustrate in Appendix E.3 that this objective does not always align with SSP. In contrast, retro-fallback is specifically designed to maximize SSP.

Mechanistically, retro-fallback most closely resembles retro* (Chen et al., 2020), which is a variant of the older AO* algorithm (Chang & Slagle, 1971; Martelli & Montanari, 1978; Nilsson, 1982; Mahanti & Bagchi, 1985). Both retro* and retro-fallback perform a bottom-up and top-down update to determine the value of each potential action, then select actions greedily. In fact, retro-fallback's updates have cost-minimization interpretation, presented in Appendix C.1.4. The key difference between the algorithms is the node selection step: retro* considers just a single cost for each node, while retro-fallback aggregates over a vector of samples to directly optimize SSP.

Lastly, we briefly comment on several research topics which are only tangentially related (deferring fuller coverage to Appendix F). Works proposing search heuristics for retrosynthesis search algorithms (F.1) complement rather than compete with our work: such heuristics could also be applied to retro-fallback. Generative models to produce synthesis plans (F.2) effectively also function as heuristics. Methods to predict individual chemical reactions are sometimes also referred to as "retrosynthesis models" (F.3), but solve a different problem than multi-step synthesis. Finally, other works have considered generally planning in stochastic graphs (F.5), but typically in a scenario where the agent is *embedded* in the graph.

## 6 EXPERIMENTS

In this section we evaluate retro-fallback experimentally. The key question we seek to answer is whether retro-fallback does indeed maximize SSP more effectively than existing algorithms. We present additional results and explain details of the setup experimental in Appendix G.

### 6.1 EXPERIMENT SETUP

We have based our experiment design on the USPTO benchmark from Chen et al. (2020), which has been widely used to evaluate multi-step retrosynthesis algorithms. However, because this benchmark does not include a feasibility or buyability model we have made some adaptations to make it suitable for our problem setting. Importantly, because we do not know what the "best" feasibility model is, we instead test *multiple* feasibility models in the hope that the conclusions of our experiments could potentially generalize to future, more advanced feasibility models. We summarize the setup below and refer the reader to Appendix G.1 for further details.

We base all of our feasibility models on the pre-trained template classifier from Chen et al. (2020) restricted to the top-50 templates. We vary our feasibility model across two axes: the *marginal* fea-

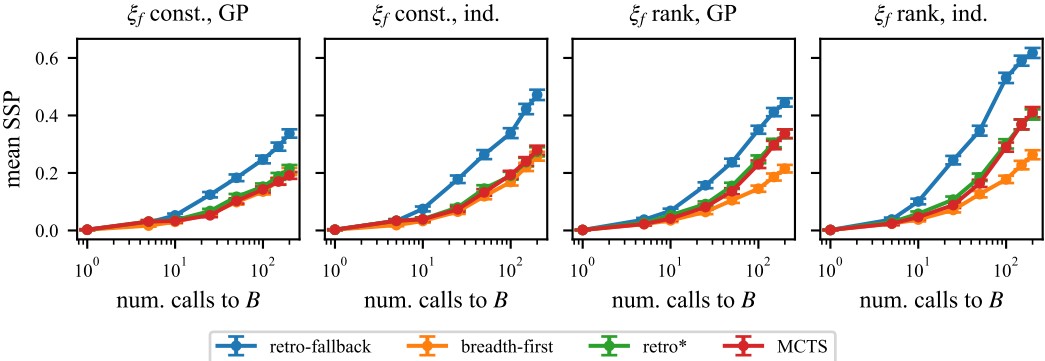

Figure 2: Mean SSP across all 190 test molecules vs. time using the *SA score* heuristic. 3 trials are done for each molecule. Solid lines are sample means (averaged across molecules), and error bars represent standard errors. "ind." means "independent".

sibility assigned to each reaction and the *correlation* between feasibility outcomes. Marginally, we consider a constant value of $0.5$, and a value which starts at $0.75$ and decreases with the rank of the reaction in the template classifier's output. For correlations, we consider all outcomes being independent or determined by a latent GP model which positively correlates similar reactions. Details of these models are given in Appendix G.1.2. Analogous to Chen et al. (2020), we create a buyability model based on the eMolecules library which designates only chemicals shipped within 10 business days as 100% buyable. See Appendix G.1.3 for details.

We compare retro-fallback to breadth-first search (an uninformed search algorithm) and the heuristic-guided algorithms retro* (Chen et al., 2020) and MCTS (Segler et al., 2018; Genheden et al., 2020; Coley et al., 2019b). All algorithms were implemented using the SYNTHESEUS library (Maziarz et al., 2023) and run with a fixed budget of calls to $B$. MCTS and retro* were configured to maximize SSP by replacing costs or rewards from the backward reaction model $B$ with quantities derived from $\xi_f$ and $\xi_b$ (see Appendices E.2 and G.1.5 for details). However, the presence of heuristics makes comparing algorithms difficult. Because the choice of heuristic will strongly influence an algorithm's behaviour, we tried to use similar heuristics for all algorithms to ensure a meaningful comparison. Specifically, we tested an *optimistic* heuristic (which gives the best possible value for each frontier node) and a heuristic based on the synthetic accessibility (SA) score (Ertl & Schuffenhauer, 2009), which has been shown to be a good heuristic for retrosynthesis in practice despite its simplicity (Skoraczyński et al., 2023). The SA score heuristic was minimally adapted for each algorithm to roughly have the same interpretation (see Appendix G.1.6 for details).

We tested all algorithms on the set of 190 "hard" molecules from Chen et al. (2020), which do not have straightforward synthesis plans. Our primary evaluation metric is the SSP values estimated with $k = 10\,000$ samples, averaged over all test molecules.

## 6.2 HOW EFFECTIVE IS RETRO-FALLBACK AT MAXIMIZING SSP?

Figure 2 plots the average SSP for all test molecules as a function of the number of calls to the reaction model $B$ using the SA score heuristic. Retro-fallback clearly outperforms the other algorithms in all scenarios by a significant margin. The difference is particularly large for the feasibility models with no correlations between reactions ("ind."). We suspect this is because the reaction model $B$ tends to output many similar reactions, which can be used to form backup plans when feasibility outcomes are independent. Retro-fallback will naturally be steered towards these plans. However, when GP-induced correlations are introduced, these backup plans disappear (or become less effective), since similar reactions will likely both be feasible or both be infeasible. The same trends are visible when using the optimistic heuristic (Figure G.4) and on a test set of easier molecules (Figure G.5) Overall, this result shows us what we expect: that retro-fallback maximizes the metric it was specifically designed to maximize more effectively than baseline algorithms.

We investigate the origin of these performance differences in Appendix G.2.1 by plotting SSP over time for a small selection of molecules (repeated over several trials). It appears that, rather than retro-

fallback being consistently a little bit better, the performance gap is driven by a larger difference for a small number of molecules. This is actually not surprising: the advantage of different approaches will vary depending on the graph, and for some graphs finding individual feasible plans is probably a promising strategy.

A natural follow-up question is whether retro-fallback also performs well by metrics other than SSP. In Figures G.8–G.10 we plot the highest success probability of any *individual* synthesis plan found, plus two metrics frequently used by previous papers: the fraction of molecules with *any* synthesis plan (called "fraction solved" in prior works) and the length of the shortest synthesis plan found (a proxy for quality). The SSP of the single best plan is generally similar for all algorithms. This suggests that in general all algorithms find similar "best" plans, and retro-fallback's extra success comes from finding more effective "backup" plans. Retro-fallback seems slightly better than other algorithms in terms of fraction solved and similar to other algorithms in terms of shortest plan length (although retro* is better in some cases). Finally, Appendix G.2.3 shows that retro-fallback is able to find synthesis plans which use the same starting molecules as real-world syntheses: a metric proposed by Liu et al. (2023b). Overall, these results suggest that retro-fallback is also an effective search algorithm if metrics from past papers which do not account for uncertainty are used.

## 6.3 SPEED AND VARIABILITY OF RETRO-FALLBACK

First we consider the speed of retro-fallback. Retro-fallback requires calculating s, $\psi$, and $\rho$ for every node at every iteration. The complexity of this calculation could scale linearly with the number of nodes in the graph (which we denote $|\mathcal{G}'|$), or potentially sub-linearly if the $s/\psi/\rho$ values for many nodes do not change every iteration. Therefore, from this step we would expect a time complexity which is between linear and quadratic in $|\mathcal{G}'|$. However, retro-fallback also requires sampling $f$ and $b$ for all nodes created during an expansion: a process which will scale as $\mathcal{O}(1)$ for independent models and $\mathcal{O}(|\mathcal{G}'|^2)$ for GP-correlated models. This yields an overall $\mathcal{O}(|\mathcal{G}'|)$–$\mathcal{O}(|\mathcal{G}'|^3)$ complexity from the sampling step. Figure G.12 plots the empirical scaling for the experiments from the previous section, and suggests an overall scaling between $\mathcal{O}(|\mathcal{G}'|^{1.1})$–$\mathcal{O}(|\mathcal{G}'|^{1.8})$, with considerable variation between different feasibility models and heuristics.

To study the effect of the number of samples $k$ from $\xi_f$ and $\xi_b$, we run retro-fallback 10 times on a sub-sample of 25 molecules with a variety of different sample sizes. Figure G.13 shows that as $k$ decreases, the mean SSP value achieved by retro-fallback decreases and the variance of SSP increases. This is not surprising, since when the number of samples is small the internal estimates of SSP used by retro-fallback deviate more from their expected values, enabling suboptimal decisions. Empirically, $k > 100$ seems sufficient (minimal further improvement is seen for higher $k$).

## 7 DISCUSSION, LIMITATIONS, AND FUTURE WORK

In this paper we reformulated retrosynthesis using stochastic processes, presented a novel evaluation metric called "successful synthesis probability" (SSP), and proposed a novel algorithm called retro-fallback which greedily maximizes SSP. In our experiments, retro-fallback was more effective at maximizing SSP than previously-proposed algorithms.

Our work has some important limitations. Conceptually, chemists may also care about the length or quality of synthesis plans, and may only be willing to consider a limited number of backup plans. These considerations do not fit into our formalism. Practically, retro-fallback is slower than other algorithms and may not scale as well. We discuss these limitations further in Appendix H.

The most important direction for future work is creating better models of reaction feasibility, as without high-quality models the estimates of SSP are not meaningful. We see collaborations with domain experts as the best route to achieve this. Since retro-fallback uses a search heuristic, learning this heuristic using the results of past searches ("self-play") would likely improve performance. We elaborate on other potential directions for future work in Appendix I.

Overall, even though retro-fallback is far from perfect, we believe that modelling uncertainty about reaction outcomes is at least a step in the right direction, and hope it inspires further work in this area.

## ETHICS

Our work is foundational algorithm development and we do not see any direct ethical implications. The most likely use case for our algorithm is to automate the production of synthesis plans in drug discovery, which we hope can aid the development of new medicines. We acknowledge the possibility that such algorithms could be used by bad actors to develop harmful chemicals, but do not see this as a probable outcome: countless harmful chemicals already exist and can be readily obtained. It is therefore hard to imagine why bad actors would expend significant effort to develop new harmful chemicals with complicated syntheses.

## REPRODUCIBILITY

We aim for a high standard of reproducibility in this work. We explicitly state our proposed algorithm in the paper (Algorithm 1) and dedicate Appendix C to discussing its minor (but still important) details, including guidance for future implementations (C.5). Proofs of all theorems are given in Appendix D. The experimental setup is described in more detail in Appendix G (including hyperparameters, etc). Code to reproduce all experiments[6] is available at:
`https://github.com/AustinT/retro-fallback-iclr24`.

Our code was thoroughly tested with unit tests and builds on libraries which are widely-used, minimizing the chance that our results are corrupted by software errors. We include the results generated by our code in `json` format, and also include code to read the results and reproduce the plots[7] from the paper. The inclusion of raw data will freely allow future researchers to perform alternative analyses.

Note that this paper will be kept updated at `https://arxiv.org/abs/2310.09270`.

## AUTHOR CONTRIBUTIONS

The original idea of SSP was proposed by Sarah and jointly developed by Sarah, Austin, Krzysztof, and Marwin. Sarah and Austin jointly developed an initial version of retro-fallback for AND/OR trees. Sarah originally proposed an algorithm using samples in a different context. Austin adapted these two algorithms to yield the version of retro-fallback proposed in this paper. Krzysztof proposed and proved Theorem 3.1. Writing was done collaboratively but mostly by Austin. All code was written by Austin with helpful code review from Krzysztof. Marwin and José Miguel advised the project. Marwin in particular provided helpful feedback about MCTS estimated feasibility of chemical reactions from the model. José Miguel provided extensive feedback on the algorithm details and the clarity of writing.

## ACKNOWLEDGMENTS

Thanks to Katie Collins for proofreading the manuscript and providing helpful feedback. Austin Tripp acknowledges funding via a C T Taylor Cambridge International Scholarship and the Canadian Centennial Scholarship Fund. José Miguel Hernández-Lobato acknowledges support from a Turing AI Fellowship under grant EP/V023756/1.

Austin is grateful for the affordable meals (with generous portion sizes) from Queens' College Cambridge which greatly expedited the creation of this manuscript.

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

# Appendix

## Table of Contents

## A  SUMMARY OF NOTATION

Although we endeavoured to introduce all notation in the main text of the paper in the section where it is first used, we re-state the notation here for clarity.

### General Math

| | |
|---|---|
| $2^S$ | Power set of set $S$ (set of all subsets of $S$) |
| $\mathbb{P}[\text{event}]$ | Probability of an event |
| $\mathbf{1}_{\text{event}}$ | Indicator function: 1 if "event" is True, otherwise 0 |
| $\mathcal{O}(N^p)$ | Big-O notation (describing scaling of an algorithm) |
| $\tilde{\mathcal{O}}(N^p)$ | Big-O notation, omitting poly-logarithmic factors (e.g. $\mathcal{O}(N \log N)$ is equivalent to $\tilde{\mathcal{O}}(N)$) |

### Molecules and reactions

| | |
|---|---|
| $m$ | a molecule |
| $r$ | a (single-product) reaction |
| $\mathcal{M}$ | space of molecules |
| $\mathcal{R}$ | space of (single-product) reactions |
| $\mathcal{I}$ | Inventory of buyable molecules |
| $B$ | backward reaction model $\mathcal{M} \mapsto 2^{\mathcal{R}}$ |
| $\rightarrow$ | Forward reaction arrow (e.g. $A + B \rightarrow C$) |
| $\Rightarrow$ | Backward reaction arrow (e.g. $C \Rightarrow A + B$) |

### Search Graphs

| | |
|---|---|
| $m_\star$ | target molecule to be synthesized |
| $\mathcal{G}$ | *implicit* search graph with molecule (OR) nodes in $\mathcal{M}$ and reaction (AND) nodes in $\mathcal{R}$ |
| $\mathcal{G}'$ | *explicit* graph stored and expanded for search. $\mathcal{G}' \subseteq \mathcal{G}$ |
| $Pa_{\mathcal{G}'}(x)$ | The parents of molecule or reaction $x$ in $\mathcal{G}'$ |
| $Ch_{\mathcal{G}'}(x)$ | The children of molecule or reaction $x$ in $\mathcal{G}'$ |
| $T$ | A synthesis plan in $\mathcal{G}$ or $\mathcal{G}'$ (conditions in Appendix B.2) |
| $\mathcal{P}_m(\mathcal{G}')$ | set of all synthesis plans in $\mathcal{G}'$ that produce molecule $m$ |
| $\mathcal{P}_*(\mathcal{G}')$ | set of all synthesis plans in $\mathcal{G}'$ producing *any* molecule |

### Feasibility and Buyability

| | |
|---|---|
| $f$ | Feasible function (assigns whether a reaction is feasible) |
| $b$ | Buyable function (assigns whether a molecule is buyable) |
| $\xi_f$ | feasibility stochastic process (distribution over $f$) |
| $\xi_b$ | buyability stochastic process (distribution over $b$) |
| $\sigma(T; f, b)$ | success of an individual synthesis plan $T$ (equation 1) |
| $\text{SSP}(\mathcal{T}; \xi_f, \xi_b)$ | Successful synthesis probability: probability that any synthesis plan $T \in \mathcal{T}$ is successful (equation 2) |
| $\text{s}(m; \mathcal{G}', f, b)$ | Whether a molecule is synthesizable using reactions/starting molecules in $\mathcal{G}'$, with feasible/buyable outcomes given by $f, b$. Takes values in $\{0, 1\}$. Defined in equations 4–5 |
| $\text{s}(m)$ | Shorthand for $\text{s}(m; \mathcal{G}', f, b)$ when $\mathcal{G}', f, b$ are clear from context. |

### Retro-fallback

| | |
|---|---|
| $h$ | Search heuristic function $\mathcal{M} \mapsto [0, 1]$ |
| $e_T$ | random expansion function (equation 7) |
| $\bar{\sigma}'(T; \xi_f, \xi_b, h)$ | estimated expected success of synthesis plan $T$ if all its frontier nodes are expanded (equation 8) |
| $\tilde{\rho}(n; \mathcal{G}', f, b, h)$ | $\bar{\sigma}'$ value of best synthesis plan for $m_\star$ including node $n$ (equation 10) |
| $\psi\rho(n; \mathcal{G}', f, b, h)$ | $\bar{\sigma}'$ value of best synthesis plan including node $n$ (equation 11) |
| $\psi(m; \mathcal{G}', f, b, h)$ | Analytic solution for $\tilde{\psi}$ under independence assumption. Defined in equations 12–13 |
| $\rho(m; \mathcal{G}', f, b, h)$ | Analytic solution for $\tilde{\rho}$ under independence assumption. Defined in equations 14–15 |
| $\alpha(m; \mathcal{G}', \xi_f, \xi_b, h)$ | expected improvement in SSP for expanding a frontier node (equation 16). |

We also use the following mathematical conventions throughout the paper:

- $\log 0 = -\infty$
- $\max_{x \in \emptyset} f(x) = -\infty$ (the maximum of an empty set is always $-\infty$)

## B  DETAILS OF SEARCH GRAPHS IN RETROSYNTHESIS

Here we provide details on the type of search graphs considered in this paper. Section 2 introduced *AND/OR graphs*. These are defined more formally in section B.1. Section 2 also informally introduced *synthesis plans* without giving a proper definition. Section B.2 provides a more precise introduction. Finally, AND/OR graphs are not the only kind of graph in retrosynthesis: MCTS typically uses an OR graph, which is introduced in section B.3.

### B.1  AND/OR GRAPHS

**Definition**  An AND/OR graph is a directed graph containing two types of nodes: AND nodes (corresponding to reactions) and OR nodes (corresponding to molecules). The name "AND/OR" originates from general literature on search. "AND" suggests that *all* child nodes must be "solved" for the node to be considered "solved", while "OR" suggests that *at least* one child node must be "solved" for the node to be considered "solved." AND nodes correspond to reactions because a reaction requires *all* input molecules, while OR nodes correspond to molecules because a molecule can be made from *any one* reaction. The connectivity of AND/OR graphs satisfies the following properties:

1. Edges only exist between molecules and reactions producing them, and between reactions and their reactant molecules. This makes the graph *bipartite*.

2. All reactions have at least one reactant (you cannot make something from nothing).

3. All reaction nodes *must* be connected to their product and reaction molecule nodes.

4. No molecule appears as both the product and a reactant in a single reaction. This implies that at most a single directed edge will exist between any two nodes.

5. All molecules are the product of a finite (and bounded) number of reactions, and all reactions have a finite (and bounded) number of molecules. This means all nodes in and AND/OR graph will always have a finite number of children.

6. Reactions contain exactly *one* product molecule. Although this may seem restrictive, the reaction $A + B \rightarrow C + D$ could simply be encoded as two reactions: $A + B \rightarrow C$ and $A + B \rightarrow D$.

**Reaction cycles: graph and tree formulations**  Reactions can form cycles (e.g. $A \Rightarrow B \Rightarrow A \Rightarrow \ldots$). There are two general ways of handling such cycles, illustrated in Figure B.1. One is to simply allow the graph to have cycles (Figure B.1a). The second is to "unroll" all cycles into a tree (Figure B.1b). Both of these ways are legitimate. Importantly however, in the tree version of an AND/OR graph *molecules and reactions may not be unique*: i.e. a molecule or reaction may occur multiple times in the graph. Because of this, in general, we assume that all functions in this paper which act on the space of molecules or reactions (e.g. s($\cdot$), $\psi(\cdot)$) really act on the space of molecule and reaction *nodes*. For example, this means that the same molecule may have different $\psi$ values at different locations in the graph.

**Implicit Search Graph**  In section 2 we stated that the implicit search graph $\mathcal{G}$ is defined by the backward reaction model. This can be either a (possibly cyclic) graph or a tree (as described above). In either case, the nodes in $\mathcal{G}$ are defined in the following way:

1. $\mathcal{G}$ contains the target molecule $m_\star$.

2. If a molecule $m \in \mathcal{G}$, then all reactions in $B(m)$ are also in $\mathcal{G}$.

3. If $\mathcal{G}$ contains the reaction $r$ (with product $m$), then $r \in B(m)$ (i.e. $r$ is the output of the backward model).

$\mathcal{G}$ can also be defined constructively by initializing $\mathcal{G}_0 \leftarrow \{m_\star\}$, and $\mathcal{G}_{i+1}$ is produced by using $B(m)$ to add reactions to add leaf nodes in $\mathcal{G}_i$. $\mathcal{G}$ is the product of repeating this infinitely many times (essentially $\mathcal{G}_\infty$). Note that by this definition, $\mathcal{G}$ could very well be an infinite graph,[8] especially if

---

[8]Hence the limit $\lim_{i \to \infty} \mathcal{G}_i$ is not well-defined.

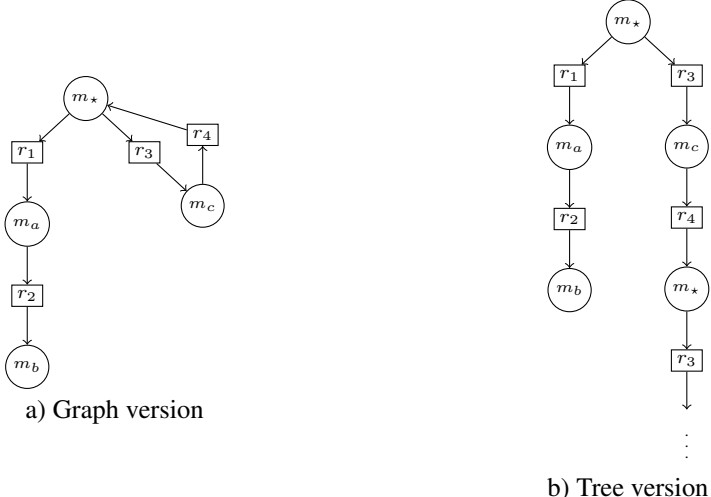

a) Graph version

b) Tree version

Figure B.1: Graph and tree representation of the reaction set $m_\star \Rightarrow m_a$ ($r_1$), $m_a \Rightarrow m_b$ ($r_2$), $m_\star \Rightarrow m_c$ ($r_3$), and $m_c \Rightarrow m_\star$ ($r_4$).

the tree construction is used (the presence of a single cycle will make $\mathcal{G}$ infinitely deep). This procedure guarantees the following properties of $\mathcal{G}$:

1. $\mathcal{G}$ is (weakly) connected. Every node in $\mathcal{G}$ is reachable from the target molecule $m_\star$ by following directed edges.

2. Only molecule nodes will have no children (reaction nodes always include their children).

**Explicit Search Graph** The explicit search graph $\mathcal{G}' \subseteq \mathcal{G}$ is a subgraph of $\mathcal{G}$ which is stored explicitly during search. $\mathcal{G}'$ will satisfy the following properties:

1. $\mathcal{G}'$ will always include the target molecule $m_\star$.

2. $\mathcal{G}'$ always contains a finite number of nodes and edges (even if $\mathcal{G}$ is infinite).

3. If two nodes $n_1, n_2 \in \mathcal{G}'$, $\mathcal{G}'$ will contain an edge between $n_1$ and $n_2$ *if any only if* $\mathcal{G}$ contains an edge between $n_1$ and $n_2$. This means $\mathcal{G}'$ will not "drop" any edges originally present in $\mathcal{G}$.

4. For a node $n \in \mathcal{G}'$, if $Ch_{\mathcal{G}'}(n) \neq \emptyset$, then no further children can be added to $n$. This means that nodes are only expanded once. Typically $Ch_{\mathcal{G}'}(n) = Ch_{\mathcal{G}}(n)$, although some children may potentially be excluded.

5. If a reaction $r \in \mathcal{G}'$, then $Ch_{\mathcal{G}'}(r) = Ch_{\mathcal{G}}(r)$. This means that reaction nodes will *always* be expanded. Expansion of a molecule node $m$ therefore involves adding all reactions in $B(m)$ and also adding the reactant molecules for all those reactions. This also means that all leaf nodes (i.e. childless nodes) in $\mathcal{G}'$ will be molecule nodes.

6. The conditions above imply that $\mathcal{G}'$ will also be weakly connected, with every node reachable from $m_\star$.

**Frontier** The *frontier* of the implicit graph $\mathcal{G}'$ is the set of nodes whose children in $\mathcal{G}$ have not been added to $\mathcal{G}'$: i.e. they are non-expanded. The frontier is denoted by $\mathcal{F}(\mathcal{G}')$. Note that the frontier of the implicit search graph $\mathcal{G}$ is empty by definition: the graph is "maximally expanded" (and therefore possibly infinite in size). However, there is one ambiguous case which must be dealt with in practice: molecules $m$ which are not the product of any reaction (i.e. $B(m) = \emptyset$). These nodes have no children in $\mathcal{G}$, and therefore expanding them will not add any children. In practice, because $B(m)$ is only calculated upon expansion, we treat these molecules as on the frontier when they are first added to the graph, and then remove them from the frontier when they are expanded. This has two practical implications:

1. $\mathcal{F}(\mathcal{G}')$ is not just the set of leaf nodes in $\mathcal{G}'$: there may be some leaf nodes not on the frontier because they have no child reactions.

2. The frontier therefore cannot be determined simply by looking at the connectivity of $\mathcal{G}'$. Our code therefore explicitly stores whether each leaf node is on the frontier. However, $\mathcal{F}(\mathcal{G}')$ will always be a *subset* of all leaf nodes in $\mathcal{G}'$.

## B.2 SYNTHESIS PLANS

Synthesis plans were defined informally in section 2. More formally, a synthesis plan $T$ is a subgraph of $\mathcal{G}$ (either as a graph or a tree) satisfying the following properties:

1. $T$ is a weakly connected subgraph of $\mathcal{G}$.

2. If two nodes $n_1, n_2 \in T$ share an edge in $\mathcal{G}$, this edge will also be present in $T$.

3. $T$ is acyclic. This means there will always be a meaningful order to "execute" $T$.

4. Every molecule $m \in T$ has *at most* one child reaction (which will also be in $\mathcal{G}$). This means that there is a unique way to make each molecule in $T$. Molecules with no child reactions form the frontier $\mathcal{F}(T)$.[9]

5. If a reaction $r \in T$, then all its children must all be in $T$ (i.e. $Ch_{\mathcal{G}}(r) \subseteq T$).

6. $T$ contains a special "root molecule" $m_\dagger$, which can be thought of as the "product" of the synthesis plan. $m_\dagger$ has no parents in $T$ (hence the plan "terminates" at $m_\dagger$).

Note in particular that each molecule $m$ has a valid synthesis plan $\{m\}$, i.e. a singleton synthesis plan. This is not a mistake; such synthesis plans would be a prudent choice for buyable molecules (and would correspond to just buying the molecule). Jiménez & Torras (2000) contains a more thorough discussion of what constitutes a synthesis plan (although using different terminology).

$\mathcal{P}_m(\mathcal{G}')$ denotes the set of all synthesis plans in $\mathcal{G}'$ whose product is $m$. $\mathcal{P}_m(\mathcal{G}')$ can be enumerated recursively, for example using Algorithm 2.[10] Shibukawa et al. (2020) provides another method to do this.

---

**Algorithm 2** Simple algorithm to enumerate all synthesis plans.

---

**Require:** Explicit graph $\mathcal{G}'$, product molecule $m_\dagger$.
1: $S \leftarrow \emptyset$        {Initialize the set of plans to be returned}
2: $S \leftarrow S \cup \{m_\dagger\}$        {Include the "singleton" plan}
3: **for** $r \in Ch_{\mathcal{G}'}(m_\dagger)$ **do**
4:      $m_1, \ldots, m_\ell \leftarrow Ch_{\mathcal{G}'}(r)$        {Extract reactant molecules}
5:      Recursively compute $\mathcal{P}_{m_1}(\mathcal{G}'), \ldots, \mathcal{P}_{m_\ell}(\mathcal{G}')$
6:      **for** every combination of $T_1 \in \mathcal{P}_{m_1}(\mathcal{G}'), \ldots, T_\ell \in \mathcal{P}_{m_\ell}(\mathcal{G}')$ **do**
7:         Form the subgraph $\tilde{T} \leftarrow \{m_\dagger, r\} \cup \left[\bigcup_i T_i\right]$
8:         **if** $\tilde{T}$ is acyclic **then**
9:           $S \leftarrow S \cup \{\tilde{T}\}$        {$\tilde{T}$ is a valid synthesis plan}
10:        **end if**
11:      **end for**
12: **end for**
13: **return** $S$

---

---

[9]This is almost identical to the definition of the frontier for AND/OR graphs from section B.1, except all leaf nodes are considered frontier nodes (i.e. there no special treatment for leaf nodes with no children in $\mathcal{G}$).

[10]However, this particular algorithm as stated may not terminate unless care is taken to avoid subgraphs which include $m_\dagger$ when recursively computing synthesis plans for reactant molecules.

## B.3 OR GRAPHS

AND/OR graphs are not the only type of graph in retrosynthesis. Other works, notably MCTS (Segler et al., 2018), use OR graphs: a graph type where each node corresponds to a *set* of molecules. Synthesis plans in such graphs are simple paths. An example of an OR graph (specifically a tree) is given in Figure B.2.

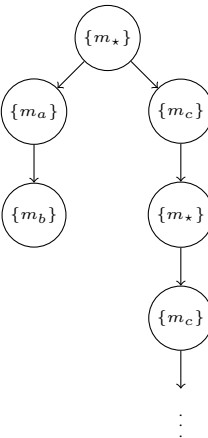

Figure B.2: OR graph for same set of reactions as Figure B.1.

## C  FURTHER DETAILS OF RETRO-FALLBACK

### C.1  DETAILS OF s($\cdot$), $\psi(\cdot)$ AND $\rho(\cdot)$

#### C.1.1  MNEMONICS

The names of s, $\psi$, and $\rho$ were chosen in an attempt to make them as intuitive as possible:

- s was chosen to represent "success".
- $\psi$ (**psi**) was chosen from the phrase "**p**robability of **s**uccess," as it can more or less be interpreted as the probability that a node *could* be successful upon expansion.
- $\rho$ was chosen as the probability of success of the **ro**ot molecule of the search graph, which is another term for the target molecule $m_\star$.

We hope the reader finds these mnemonics useful.

#### C.1.2  RELATIONSHIPS BETWEEN s($\cdot$), $\psi(\cdot)$, AND $\rho(\cdot)$

Note that, for all $T \in \mathcal{P}_*(\mathcal{G}')$ and $e_T$

$$\underbrace{\sigma(T; f, b)}_{\text{equation 1}} \leq \underbrace{\sigma'(T; f, b, e_T)}_{\text{equation 7}} . \tag{17}$$

Taking an expectation with respect to $e_T$ on both sides gives

$$\sigma(T; f, b) \leq \bar{\sigma}'(T; f, b, h) . \tag{18}$$

Noting the relationship between s($\cdot$) and $\psi(\cdot)$, this implies for all nodes $n \in \mathcal{G}'$ and heuristics $h$

$$\text{s}(n; \mathcal{G}', f, b) \leq \psi(n; \mathcal{G}', f, b, h) . \tag{19}$$

Finally, since the set of synthesis plans considered for $\tilde{\rho}$ is a subset of the plans considered for $\tilde{\psi}$, it must be that

$$\rho(n; \mathcal{G}', f, b, h) \leq \psi(n; \mathcal{G}', f, b, h) . \tag{20}$$

#### C.1.3  ANALYTIC SOLUTIONS FOR $\psi(\cdot)$ AND $\rho(\cdot)$

First, we state more formally the assumptions under which these analytic solutions hold.

**Assumption C.1** (No synthesis plan uses the same molecule twice). *Assume for all $T \in \mathcal{P}_*(\mathcal{G}')$, if $r_1, r_2 \in T$ are distinct reactions in $T$ (i.e. $r_1 \neq r_2$), then $Ch_T(r_1) \cap Ch_T(r_2) = \emptyset$ (i.e. $r_1$ and $r_2$ share no children in $T$).*

*Furthermore, if $m_1, m_2 \in T$ are distinct molecule nodes, then $m_1$ and $m_2$ are themselves distinct molecules.*[11]

**Corollary C.2.** *Under assumption C.1, all synthesis plans $T \in \mathcal{P}_*(\mathcal{G}')$ are* trees

*Proof.* Synthesis plans were explicitly defined to be acyclic in Appendix B.2, so it is sufficient to show that no two nodes in an arbitrary $T$ share the same parent. Two reaction nodes cannot share the same parent (synthesis plans were defined as having at most one child for each molecule node), and if a molecule node has two reaction parents assumption C.1 would be violated. $\square$

Next, we state a useful lemma for the upcoming proofs about $\psi$ and $\rho$. This lemma and the following propositions all assume a fixed (and arbitrary) graph $\mathcal{G}'$ and functions $f, b, h$.

**Lemma C.3.** *If $T_1$ and $T_2$ are synthesis plans, and $\mathcal{F}(T_1) \subseteq \mathcal{F}(T_2)$ (i.e. all frontier nodes in $T_1$ are in $T_2$), then $\bar{\sigma}'(T_1; f, b, h) \leq \bar{\sigma}'(T_2; f, b, h)$.*

---

[11]This additional statement is necessary for tree-structured search graphs, wherein the same molecule may occur in several different nodes.

*Proof.* Equation 8 imposes a condition on *all* frontier nodes for a synthesis plan, so clearly for all $e_T$,

$$\sigma'(T_1; f, b, e_T) \leq \sigma'(T_2; f, b, e_T) \, .$$

Since this equality holds for all $e_T$, it will also hold in expectation over $e_T$, which is the definition of $\bar{\sigma}'$ (equation 9). □

Now we prove the statements from the main text. First, we consider $\tilde{\psi}(\cdot)$ (equation 11).

**Proposition C.4.** *Under assumption C.1, $\tilde{\psi}(\cdot; \mathcal{G}', f, b, h)$ satisfies equations 12–13.*

*Proof.* Nodes in $\mathcal{G}'$ can be partitioned into the following sets, for which the result is shown separately.

1. Frontier molecules $m \in \mathcal{F}(\mathcal{G}')$: the synthesis plan $T_m = \{m\}$ has $\bar{\sigma}'(T_m; f, b, h) = \max[b(m), h(m)]$ (from equation 8 and equation 7). All other synthesis plans $T'$ including $m$ will include it as a leaf node, and by lemma C.3 will have $\bar{\sigma}'(T'; f, b, h) \leq \bar{\sigma}'(T_m; f, b, h)$. Therefore $T_m$ maximizes $\bar{\sigma}'$ (although it might not be unique), justifying the first case of equation 12.

2. Non-frontier molecules $m \notin \mathcal{F}(\mathcal{G}')$: first, if $m$ is a non-frontier node with no children, then all synthesis plans including $m$ will terminate in $m$, giving $\psi(m) = b(m)$. Equation 12 matches this (because we use the convention $\max \emptyset = -\infty$).

   Otherwise, first consider the case where $b(m) = 1$. Clearly $T_m = \{m\}$ maximizes equation 11, so $\psi(m; \mathcal{G}', f, b, h) = b(m) = 1$.

   Otherwise, consider $b(m) = 0$. In this case, $\bar{\sigma}'(T_m; f, b, h) = 0$, so all other synthesis plans will be equal to or better than $T_m$. Therefore we look to $m$'s child reactions. Synthesis plans containing one of $m$'s child reactions will necessarily contain $m$, making $\psi(m) = \max_{r \in Ch_{\mathcal{G}'}(m)} \psi(r)$: exactly what is in equation 12.

3. Reaction nodes $r$: Because of assumption C.1 and the fact that $e_T$ (equation 7) is independent for every node, the optimal synthesis plan including $r$ will be formed by combining the optimal synthesis plans for all reactions $m \in Ch_{\mathcal{G}'}(r)$. Due to independence, the probability of them all being successful is simply the product of each synthesis plan being successful, justifying equation 13.

□

Finally, we consider $\tilde{\rho}$ (equation 10).

**Proposition C.5.** *Under assumption C.1, $\tilde{\rho}(\cdot; \mathcal{G}', f, b, h)$ satisfies equations 14–15.*

*Proof.* First, note that from our definition of AND/OR graphs in section B.1 all nodes in $\mathcal{G}'$ will be included on at least one synthesis plan which produces $m_\star$.

We partition nodes in $\mathcal{G}'$ into the following sets and provide a separate proof for each set.

1. The target molecule $m_\star$: first, any synthesis plan $T$ including $m_\star$ as an intermediate molecule (i.e. not the product of the synthesis plan) will have $\bar{\sigma}'(T; f, b, h)$ bounded by $\bar{\sigma}'(T'; f, b, h)$ for some synthesis plan $T'$ whose final product is $m_\star$ (by lemma C.3). Therefore, and synthesis plan achieving $\psi(m_\star; \mathcal{G}', f, b, h)$ will also maximize $\tilde{\rho}$. So, $\rho(m_\star; \mathcal{G}', f, b, h) = \psi(m_\star; \mathcal{G}', f, b, h)$.

2. A non-target molecule $m$: $m$ will have at least one parent $r$. Any synthesis plan for $m_\star$ which includes $r$ must necessarily include $m$ (because synthesis plans will always contain all reactants for every reaction). Therefore, the set of synthesis plans for $m_\star$ which also contain $m$ is given by

$$\cup_{r \in Pa_{\mathcal{G}'}(m)} \{T \in \mathcal{P}_\star(\mathcal{G}') : r \in T\} \, .$$

   $\rho(m)$ will simply be the maximum of these values.

Together with the previous case, this justifies equation 14.

3. A reaction $r$. First, if $\psi(r; \mathcal{G}', f, b, h) = 0$ then clearly $\rho(r; \mathcal{G}', f, b, h) = 0$ as $\psi$ maximizes over a larger set of synthesis plans than $\rho$.

   Otherwise, let $m$ be the single parent of $r$. The synthesis plan $T$ justifying $\rho(m; \mathcal{G}, f, b, h)$ will be composed of a synthesis plan $T_m$ for $m$ (i.e. "below" $m$) and a partial synthesis plan $T \setminus T_m$ which uses $m$ as an intermediate molecule (or potentially the product molecule if $m = m_\star$). Because of the independence of different branches resulting from equation 7 and assumption C.1, $\bar{\sigma}'(T; f, b, h)$ will be the product of $\bar{\sigma}'(T_m; \mathcal{G}', f, b, h) = \psi(m; \mathcal{G}', f, b, h)$ and a term for $T \setminus T_m$. The optimal synthesis plan which *includes* $r$ can therefore be constructed by removing $T_m$ (and dividing by its probability $\psi(m; \mathcal{G}', f, b, h)$) and adding the optimal synthesis plan for $m$ which includes $r$ (with probability $\psi(r; \mathcal{G}', f, b, h)$).

   Together, these cases justify equation 15.

   $\square$

### C.1.4 COST-MINIMIZATION INTERPRETATION

The functions $s(\cdot)$, $\psi(\cdot)$, and $\rho(\cdot)$ have a cost minimization interpretation when transformed by mapping $f(\cdot) \mapsto -\log f(\cdot)$ (using the convention that $\log 0 = -\infty$). To see, this, we write the transformed equations explicitly. To simplify notation, we write:

$$\dot{b}(m) = -\log b(m) \tag{21}$$

$$\dot{f}(r) = -\log f(r) \tag{22}$$

$$\dot{h}(m) = -\log h(m) \tag{23}$$

$$\dot{s}(\cdot) = -\log s(\cdot) \tag{24}$$

$$\dot{\psi}(\cdot) = -\log \psi(\cdot) \tag{25}$$

$$\dot{\rho}(\cdot) = -\log \rho(\cdot) . \tag{26}$$

For $s(\cdot)$ (equations 4–5), we have

$$\dot{s}(m; \mathcal{G}', f, b) = \min \left[ \dot{b}(m), \min_{r \in Ch_{\mathcal{G}'}(m)} \dot{s}(r; \mathcal{G}', f, b) \right] , \tag{27}$$

$$\dot{s}(r; \mathcal{G}', f, b) = \dot{f}(r) + \sum_{m \in Ch_{\mathcal{G}'}(r)} \dot{s}(m; \mathcal{G}', f, b) . \tag{28}$$

Since $s(\cdot) \in \{0, 1\}$, this corresponds to a pseudo-cost of $0$ if a molecule is synthesizable and a cost of $+\infty$ otherwise.

For $\psi(\cdot)$ (equations 12–13) and $\rho(\cdot)$ (equations 14–15) we have:

$$\dot{\psi}(m; \mathcal{G}', f, b, h) = \begin{cases} \min \left[ \dot{b}(m), \dot{h}(m) \right] & m \in \mathcal{F}(\mathcal{G}') \\ \min \left[ \dot{b}(m), \min_{r \in Ch_{\mathcal{G}'}(m)} \dot{\psi}(r; \mathcal{G}', f, b, h) \right] & m \notin \mathcal{F}(\mathcal{G}') \end{cases} , \tag{29}$$

$$\dot{\psi}(r; \mathcal{G}', f, b, h) = \dot{f}(r) + \sum_{m \in Ch_{\mathcal{G}'}(r)} \dot{\psi}(m; \mathcal{G}', f, b, h) . \tag{30}$$

$$\dot{\rho}(m; \mathcal{G}', f, b, h) = \begin{cases} \dot{\psi}(m; \mathcal{G}', f, b, h) & m \text{ is target molecule } m_\star \\ \min_{r \in Pa_{\mathcal{G}'}(m)} \dot{\rho}(r; \mathcal{G}', f, b, h) & \text{all other } m \end{cases} , \tag{31}$$

$$\dot{\rho}(r; \mathcal{G}', f, b, h) = \begin{cases} +\infty & \dot{\psi}(r; \mathcal{G}', f, b, h) = +\infty \\ \dot{\rho}(m'; \mathcal{G}', f, b, h) \\ \quad -\dot{\psi}(m'; \mathcal{G}', f, b, h) \\ \quad +\dot{\psi}(r; \mathcal{G}', f, b, h) & \dot{\psi}(r; \mathcal{G}', f, b, h) < \infty, m' \in Pa_{\mathcal{G}'}(r) \end{cases} . \tag{32}$$

Since $\psi(\cdot)$ and $\rho(\cdot)$ have ranges $[0, 1]$, $\dot{\psi}(\cdot)$ and $\dot{\rho}(\cdot)$ have ranges $[0, +\infty]$ (i.e. explicitly including infinity).

### C.1.5 CLARIFICATION OF $rs$, $\psi$, AND $\rho$ FOR CYCLIC GRAPHS

If $\mathcal{G}'$ is an acyclic graph, the recursive definitions of s$(\cdot)$, $\psi(\cdot)$, and $\rho(\cdot)$ will always have a unique solution obtained by direct recursion (from children to parents for s$(\cdot)$ and $\psi(\cdot)$, and from parents to children for $\rho(\cdot)$). However, for cyclic graphs, such recursion is not possible and therefore these equations do not (necessarily) uniquely define the functions s$(\cdot)$, $\psi(\cdot)$, and $\rho(\cdot)$.

The minimum cost interpretation from section C.1.4 provides a resolution. Minimum cost solutions in AND/OR graphs with cycles have been extensively studied (Hvalica, 1996; Jiménez & Torras, 2000) and do indeed exist. Hvalica (1996, Lemma 1) in particular suggests that the presence of multiple solutions will happen only because of the presence of zero-cost cycles (which could be caused by a cycle of feasible reactions). To produce a unique solution, we re-define all costs to 0 to be a constant $\epsilon > 0$ (for which there will be a unique solution for $\dot{s}$, $\dot{\psi}$, and $\dot{\rho}$), and take the limit as $\epsilon \to 0$ to produce a unique solution which includes 0 costs.

### C.1.6 COMPUTING s, $\psi$, AND $\rho$

The best method of computation will depend on $\mathcal{G}'$. If $\mathcal{G}'$ is acyclic (for example, as will always be the case for tree-structured search graphs) then direct recursion is clearly the best option. This approach also has the advantage that the computation can stop early if the value of a node does not change.

In graphs with cycles, any number of algorithms to find minimum *costs* in AND/OR graphs can be applied to the minimum-cost formulations of s$(\cdot)$, $\psi(\cdot)$, and $\rho(\cdot)$ from section C.1.4 (Chakrabarti, 1994; Hvalica, 1996; Jiménez & Torras, 2000).

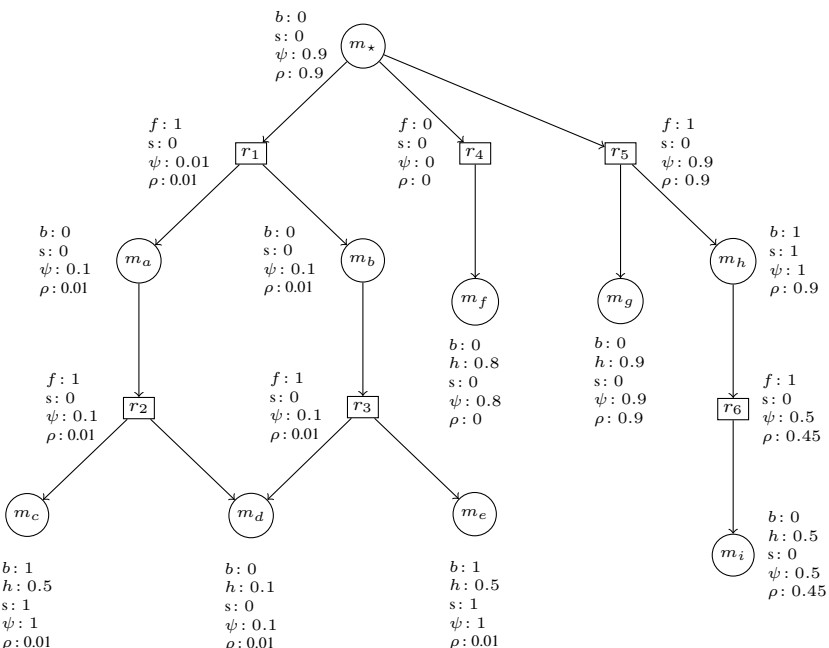

Figure C.1: A search graph $\mathcal{G}'$ with values for s, $\psi$, and $\rho$ worked out. A detailed explanation is given in Appendix C.2.

## C.2 EXAMPLE OF CALCULATING s, $\psi$ AND $\rho$

Figure C.1 presents a search graph $\mathcal{G}'$ with labels from one sample of $f, b$, and heuristic values $h$. The graph contains three branches stemming from $m_\star$, starting with $r_1$, $r_4$, and $r_5$ respectively. It is designed to highlight several interesting corner cases, which will be emphasized below. We will work through the calculations of s, $\psi$, and $\rho$ for all nodes below. To simplify notation, we will omit the dependence of these quantities on $\mathcal{G}', f, b, h$, as this is implicit.

### C.2.1 CALCULATION OF s

Note that only molecules $m_c$, $m_e$, and $m_h$ are buyable ($b(\cdot) = 1$); the rest are non-buyable. All reactions are feasible ($f(\cdot) = 1$), except for $r_4$. This means that $s(\cdot) = 1$ only for $m_c$, $m_e$, and $m_h$. Their parent reactions all have one other reactant with $s(\cdot) = 0$; therefore all other nodes in the graph have $s(\cdot) = 0$.

### C.2.2 CALCULATION OF $\psi$

Since $\psi$ depends only on a node's children, we will show the calculation bottom-up for each branch.

$r_1$ **branch:**

- **Leaf nodes:** $m_c$ and $m_e$ are buyable molecules whose heuristic values are less than 1. Therefore, $h(m_c) = h(m_d) = 0.5$ is dominated by $b(\cdot) = 1$, in the $\max$ of equation 12, yielding $\psi(\cdot) = 1$. For $m_d$ which is not buyable, $\psi(m_d) = h(m_d) = 0.1$.

- $r_2, r_3$: Both these reactions are feasible ($f(\cdot) = 1$), so $\psi(\cdot)$ will just be the product of $\psi$ values of its children, which in this case yields $\psi(r_2) = \psi(r_3) = 0.1$.

- $m_a, m_b$: Both molecules inherit $\psi$ values from their children, so $\psi(m_a) = \psi(m_b) = 0.1$.

- $r_1$: $f(r_1) = 1$, so by equation 13 $\psi(r_1) = \psi(m_a)\psi(m_b) = 0.01$. *Note however that this is an instance of double counting, because $m_d$ occurs twice in this synthesis plan. $\psi(r_1)$ calculates a success probability for $r_1$ as if $m_d$ needs to independently succeed twice, yielding a value of 0.01 instead of 0.1. Synthesis plans like this are unusual however, and*

as explained previously we simply accept that to calculate $\psi$ efficiently it may occasionally double-count certain nodes.

$r_4$ **branch:**

- $m_f$: Here, $\psi(m_f) = h(m_f) = 0.8$ (since $b(m_f) = 0$).
- $r_4$: $\psi(r_4) = f(r_4)\psi(m_f) = 0(0.8) = 0$. *Essentially, because this reaction is infeasible, $\psi$ will be 0*, since no amount of expansion below $r_4$ will make it feasible.

$r_5$ **branch:**

- $m_i$: Here, $\psi(m_i) = h(m_i) = 0.5$.
- $r_6$: $\psi(r_6) = f(r_6)\psi(m_i) = (1)(0.5) = 0.5$.
- $m_h$: This is another edge case: *a buyable molecule with child nodes*. Here, $\psi(m_h) = \max[b(m_h), \psi(r_6)] = 1$. Essentially, it inherits its $\psi$ value from $b$, not from $\psi$ values of its children.
- $m_g$: Here, $\psi(m_g) = h(m_g) = 0.9$.
- $r_5$: $\psi(r_5) = f(r_5)\psi(m_g)\psi(m_h) = (1)(0.9)(1) = 0.9$.

$m_\star$:

$$\psi(m_\star) = \max[b(m_\star), \psi(r_1), \psi(r_4), \psi(r_5)]$$
$$= \max[0, 0.01, 0, 0.9]$$
$$= 0.9$$

The optimal synthesis plan achieving this success probability is

$$\{m_\star, r_5, m_g \text{ (expanded)}, m_h \text{ (bought)}\}.$$

### C.2.3 CALCULATION OF $\rho$

Since $\rho$ depends only on a node's parents, we will show the calculation top-down.

$m_\star$: By definition in equation 14, $\rho(m_\star) = \psi(m_\star) = 0.9$.

$r_5$ **branch:** Note that the optimal synthesis plan which justifies $\psi(m_\star)$ comes from this branch, so we should expect many nodes to have $\rho(\cdot) = \psi(m_\star)$.

- $r_5$: From equation 15, $\rho(r_5) = \rho(m_\star)\frac{\psi(r_5)}{\psi(m_\star)} = 0.9\frac{0.9}{0.9} = 0.9$. As expected, this equals $\psi(m_\star)$.
- $m_g, m_h$: From equation 14 $\rho(m_g) = \rho(m_h) = \rho(r_5) = 0.9$.
- $r_6$: *This is an interesting edge case because $r_6$'s parent, $m_h$, is buyable*. From the definition of $\rho$, $\rho(r_6)$ should be a quantity which only considers synthesis plans which include $r_6$, and therefore must not buy $m_h$. We can work out:

$$\rho(r_6) = \rho(m_h)\frac{\psi(r_6)}{\psi(m_h)}$$
$$= 0.9\frac{0.5}{1}$$
$$= 0.45$$

  Intuitively this lower value makes sense: using $r_6$, which is only predicted to succeed with 50% probability, reduces the success probability of the best synthesis route by half.
- $m_i$: $\rho(m_i) = \rho(r_6) = 0.5$

$r_4$ **branch:** This branch uses an infeasible reaction, so any synthesis plan forced to contain nodes in this branch should have a success probability of 0. Indeed we see this below:

- $r_4$: $\rho(r_4) = \rho(m_\star)\frac{\psi(r_4)}{\psi(m_\star)} = 0.9\frac{0}{0.9} = 0$.

- $m_f$: $\rho(m_f) = \rho(r_4) = 0$.

$r_1$ **branch:** This branch contains just one synthesis plan with predicted success of $0.01$ (due to the double counting). Calculating $\rho$ should therefore propagate this value down to the leaf nodes. Indeed, this is what we see below:

- $r_1$: $\rho(r_1) = \rho(m_\star)\frac{\psi(r_1)}{\psi(m_\star)} = 0.9\frac{0.01}{0.9} = 0.01$.

- $m_a, m_b$: $\rho(m_a) = \rho(m_b) = \rho(r_1) = 0.01$.

- $r_2, r_3$: $\rho(r_2) = \rho(m_a)\frac{\psi(r_2)}{\psi(m_a)} = 0.01\frac{0.1}{0.1} = 0.01$ (and same for $r_3$).

- $m_c, m_d, m_e$: Note that $\rho(m_c) = \rho(r_2) = 0.01$ and $\rho(m_e) = \rho(r_3) = 0.01$. Node $m_d$ has multiple parents, but all with the same $\rho$ value, so $\rho(m_d) = \min[0.01, 0.01] = 0.01$.

### C.3 JUSTIFICATION OF NODE SELECTION

Selecting nodes for expansion based on $\rho$ values for samples $f, b$ where $s(m_\star; \mathcal{G}'f, b) = 0$ will always have the interpretation of improving SSP. Taking the expected value in this scenario is a natural choice. However, it is not the only choice. The formulas

$$\alpha_{\text{Mode}}(m; \mathcal{G}', \xi_f, \xi_b, h) = \text{Mode}_{f \sim \xi_f, b \sim \xi_b}\left[\mathbf{1}_{s(m_\star; \mathcal{G}', f, b) = 0}\left[\rho(m; \mathcal{G}', f, b, h)\right]\right] \tag{33}$$

$$\alpha_q(m; \mathcal{G}', \xi_f, \xi_b, h) = \text{Quantile}^q_{f \sim \xi_f, b \sim \xi_b}\left[\mathbf{1}_{s(m_\star; \mathcal{G}', f, b) = 0}\left[\rho(m; \mathcal{G}', f, b, h)\right]\right] \tag{34}$$

are also sensible choices. We choose the expected value mainly out of simplicity.

### C.4 REJECTED ALTERNATIVE ALGORITHMS

The first iteration of retro-fallback (*proto retro-fallback*) used a search tree instead of a search graph, and assumed that the feasibility/buyability of all reactions/molecules was *independent*. In this special case, the values of $s, \psi$ and $\rho$ can all be computed analytically using dynamic programming. However, a major weakness of this algorithm is that forcing $\mathcal{G}'$ to be a tree required duplicating some molecules and reactions in the graph (e.g. if both the reactions $A + B \Rightarrow C$ and $A + D \Rightarrow C$ are possible then the molecule $A$ and any reactions under it would be duplicated). The assumption of independence meant that the feasibility of the *same reactions* would be sampled multiple times independently, leading to "backup plans" that actually used the same reaction. In practice this was often not an issue, but it did mean that the internal estimates of $s$ used by proto retro-fallback did not have a clear relationship to true SSP. Hence we decided to proceed using samples, which provided a natural avenue to remove the independence assumption. More details about proto retro-fallback can be given upon request to the authors. An implementation of this version of retro-fallback is also available in our public code.

When designing the version of retro-fallback presented in this paper, we first considered *sampling* outcomes for frontier nodes using the heuristic function, and updating using the standard equations for $s$. This would effectively be a random heuristic, and although other algorithms use random heuristics (e.g. rollouts in MCTS) we decided that it would be an extra source of variance, and upon realizing that the expected value of such a heuristic can be computed analytically if the outcomes are assumed to be independent we proceeded to develop the equation for $\psi$. However, if in the future other researchers wish to remove the independence assumption in the heuristic then probably its outcomes would also need to be sampled.

### C.5 PRACTICAL IMPLEMENTATION DETAILS

Here we give further details of how retro-fallback can be implemented. This section is *not* a description of our specific software implementation used in this paper: that is in Appendix G.1.1. Instead, we try to give general guidance that would be applicable for alternative implementations and alleviate potential sources of confusing or ambiguities.

**Graph initialization** In Algorithm 1 we start by initializing $\mathcal{G}'$ to contain just the target molecule. Strictly speaking this is not required: any starting graph can be used (as long as it satisfies the

assumptions about AND/OR graphs in Appendix B.1, e.g. not having reactions as frontier nodes or edges between molecules and reactions that should not be connected). All that is needed is to properly initialize the variables (s, $\psi$, $\rho$) before entering the main loop.

**Samples from stochastic processes** A full sampled function from $\xi_f$ or $\xi_b$ contains outcomes for every possible molecule or reaction. In reality this could not be calculated or stored. Therefore, in practice one would only sample these functions for nodes in $\mathcal{G}'$, and sample from the *posterior* stochastic processes when new nodes are added. For processes where this is inexpensive (e.g. processes with independent outcomes) this is fast and the implementation is not important. However, when it is slow it is likely very important to use caching. For example, drawing samples from a GP posterior scales with $O(N^3)$: if $\xi_f$ or $\xi_b$ use a GP, this $O(N^3)$ operation at every step will result in an overall algorithm speed of $O(N^4)$! To avoid this, we cached a Cholesky decomposition of the GP covariance matrix at every step and used incremental updating of the Cholesky decomposition to bring the overall complexity to at most $O(N^3)$. For other stochastic processes different techniques may be applicable, but in general for non-independent stochastic processes we anticipate some form of caching may be necessary.

**Vectorized computation** Vectorized computation could also be used instead of explicit `for` loops over the sampled functions in $1, \ldots, k$, wherein values for s, $\psi$, and $\rho$ would be simultaneously updated for all samples.

**Priority Queues** In each iteration of algorithm 1 a frontier node maximizing $\alpha$ is found. In practice, this could be accelerated using a priority queue to avoid explicitly considering every frontier node in every iteration (which has a cost linear in the number of frontier nodes).

**Backward reaction model** Algorithm 1 requires a backward reaction model $B$. This is also not necessary: all that is needed is some way to decide what reactions to add to the graph. For example, if it was possible to obtain a list of reactions from $\xi_f$ whose marginal feasibility is non-zero, this could be used as a replacement for $B$.

**Tie-breaking** Algorithm 1 suggests when choosing nodes, ties should be broken arbitrarily. However, some frontier nodes may be buyable, and will only have a high $\rho$ value because they belong to the synthesis plans with other expandable nodes. Therefore, in general it would be beneficial to break ties in favour of *non-buyable* nodes, since expanding a buyable node will never produce a new synthesis plan.

**Termination conditions** Algorithm 1 uses several termination conditions, some of which may not be necessary or could be modified:

1. No nodes are left to expand. We believe this one is necessary.

2. $L$ iterations of expansion are done. This is not necessary: the algorithm alternatively terminate after a fixed wall-clock time, or simply never terminate until it is killed by the user.

3. All $s_i(m_\star)$ are 1: this is a sensible point to terminate because it means that $\alpha(m) = 0$ for all frontier nodes $m$. However, the algorithm could easily keep running past this point; it would just expand nodes arbitrarily because all nodes would have an equivalent value of $\alpha(m)$. This condition could also be loosened: for example the algorithm could terminate when $\hat{s}(m_\star) > 1 - \epsilon$ for some small $\epsilon > 0$. This is sensible if one believes that improvement beyond a certain point is redundant.

# D    PROOFS AND THEORETICAL RESULTS

This appendix contains proofs of theoretical results from the paper.

## D.1    PROOF OF THEOREM 3.1

Theorem 3.1 is a corollary of the following theorem, which we prove below.

**Theorem D.1.** *Unless $P = NP$, there does not exist an algorithm to determine whether* $\mathrm{SSP}(\mathcal{P}_{m_\star}(\mathcal{G}'); \xi_f, \xi_b) > 0$ *for arbitrary $\xi_f, \xi_b$ whose time complexity grows polynomially with the number of nodes in $\mathcal{G}'$.*

Note that Theorem D.1 is similar to but distinct from Theorem 3.1. The difference (highlighted) is that Theorem 3.1 is about the difficulty of computing SSP, while Theorem D.1 is only about the difficulty of determining whether or not SSP is 0. We now state a proof of Theorem D.1:

*Proof.* We will show a reduction from the Boolean 3-Satisfiability Problem (3-SAT) to the problem of determining whether SSP is non-zero. As 3-SAT is known to be NP-hard (Karp, 1972), this will imply the latter is also NP-hard, completing the proof.

To construct the reduction, assume an instance $I$ of 3-SAT with $n$ variables $x_1, \ldots, x_n$, and $m$ clauses $c_1, \ldots, c_m$, each $c_j$ consisting of three literals (where a literal is either a variable or its negation). We will construct an AND-OR graph $\mathcal{G}(I)$ with size $\mathcal{O}(n + m)$, alongside with distributions $\xi_f(I)$ and $\xi_b(I)$, such that the SSP in the constructed instance is non-zero if and only if $I$ is satisfiable.

In our construction we first set $\xi_f \equiv 1$, i.e. assume all reactions described below are always feasible.

We then construct a set $P$ of $2n$ potentially buyable molecules, corresponding to variables $x_i$ as well as their negations $\neg x_i$; to simplify notation, we will refer to these molecules as $x_i$ or $\neg x_i$. We then set $\xi_b(I)$ to a uniform distribution over all subsets $S \subseteq P$ such that $|S \cap \{x_i, \neg x_i\}| = 1$ for all $i$; in other words, either $x_i$ or $\neg x_i$ can be bought, but never both at the same time. Note that with this construction it is easy to support all necessary operations on $\xi_b$, such as (conditional) sampling or computing marginals.

It remains to translate $I$ to $\mathcal{G}(I)$ in a way that encodes the clauses $c_j$. We start by creating a root OR-node $r$, with a single AND-node child $r'$. Under $r'$ we build $m$ OR-node children, corresponding to clauses $c_j$; again, we refer to these nodes as $c_j$ for simplicity. Finally, for each $c_j$, we attach 3 children, corresponding to the literals in $c_j$. Intuitively these 3 children would map to three molecules from the potentially buyable set $P$, but formally the children of $c_j$ should be AND-nodes (while $P$ contains molecules, i.e. OR-nodes); however, this can be resolved by adding dummy single-reactant reaction nodes.

To see that the reduction is valid, first note that $r$ is synthesizable only if all $c_j$ are, which reflects the fact that $I$ is a binary AND of clauses $c_j$. Moreover, each $c_j$ is synthesizable if at least one of its 3 children is, which translates to at least one of the literals being satisfied. Our construction of $\xi_b$ allows any setting of variables $x_i$ as long as it's consistent with negations $\neg x_i$. Taken together, this means the SSP for $\mathcal{G}(I)$ is non-zero if and only if there exists an assignment of variables $x_i$ that satisfies $I$, and thus the reduction is sound. □

**Corollary D.2.** *If a polynomial time algorithm did exist to compute the exact value of $\mathrm{SSP}(\mathcal{P}_{m_\star}(\mathcal{G}'); \mathcal{G}', \xi_f, \xi_b)$, this algorithm would clearly also determine whether $\mathrm{SSP}(\mathcal{P}_{m_\star}(\mathcal{G}'); \mathcal{G}', \xi_f, \xi_b) > 0$ in polynomial time, violating Theorem D.1. This proves Theorem 3.1.*

## D.2    COMPUTING $\mathrm{s}(\cdot)$, $\psi(\cdot)$ AND $\rho(\cdot)$ POLYNOMIAL TIME

Here, we state provide proofs that these quantities can be computed in polynomial time. These proofs require the following assumption:

**Assumption D.3** (Bound on children size). *Assume $|Ch_{\mathcal{G}}(m)| \le K$, $|Ch_{\mathcal{G}}(m)| \le L$, $|Pa_{\mathcal{G}}(m)| \le J$ for all molecules $m$ and reactions $r$, where $J, K, L \in \mathbb{N}$ are fixed constants.*

This assumption essentially ensures that even as the retrosynthesis graph grows, the number of neighbours for each node does not grow.

Now we state our main results.

**Lemma D.4** (s and $\psi$ in polynomial time). *There exists an algorithm to calculate* $s(n; \mathcal{G}', f, b)$ *and* $\psi(n; \mathcal{G}', f, b, h)$ *for every node* $n \in \mathcal{G}'$ *whose time complexity is polynomial in the number of nodes in* $\mathcal{G}'$.

*Proof.* Section C.1.4 showed how $-\log s(\cdot)$ and $-\log \psi(\cdot)$ correspond exactly cost equations for minimum-cost synthesis plans, as studied for the AO* algorithm. (Chakrabarti, 1994) provides two algorithms, `Iterative_revise` and `REV*` whose runtime is both worst-case polynomial in the number of nodes (assuming a number of edges which does not grow more than linearly with the number of nodes, which assumption D.3 guarantees). □

The following lemma shows that, with a bit of algebraic manipulation, the same strategy can be applied to $\rho$.

**Lemma D.5.** *There exists an algorithm to calculate* $\rho(n; \mathcal{G}', f, b)$ *for every node* $n \in \mathcal{G}'$ *whose time complexity is polynomial in the number of nodes in* $\mathcal{G}'$.

*Proof.* To do this, first compute $\psi(\cdot)$ for every node (which lemma D.4 states can be done in polynomial time). Next, define the function $\eta(\cdot) = \log \frac{\psi(m_\star; \mathcal{G}', f, b, h)}{\rho(\cdot; \mathcal{G}', f, b, h)}$. This is essentially a constant transformation of $\rho(\cdot)$ at each point (since $\psi(m_\star; \mathcal{G}', f, b, h)$ is just a constant). Therefore $\eta$ has the analytical solution:

$$\eta(m; \mathcal{G}', f, b, h) = \begin{cases} 0 & m = m_\star \\ \min_{r \in Pa_{\mathcal{G}'}(m)} \eta(r; \mathcal{G}', f, b, h) & \text{all other } m \end{cases} \quad (35)$$

$$\eta(r; \mathcal{G}', f, b, h) = \begin{cases} \infty & \psi(r; \mathcal{G}', f, b, h) = 0 \\ \eta(m'; \mathcal{G}', f, b, h) + \log \frac{\psi(m'; \mathcal{G}', f, b, h)}{\psi(r; \mathcal{G}', f, b, h)} & \psi(r; \mathcal{G}', f, b, h) < \infty, m' \in Pa_{\mathcal{G}'}(r) \,. \end{cases} \quad (36)$$

Now, define the graph $\tilde{\mathcal{G}}'$ by flipping the direction of all edges in $\mathcal{G}'$ (so all parents become children, and vice versa). $\eta$ can be re-written as:

$$\eta(m; \tilde{\mathcal{G}}', f, b, h) = \begin{cases} 0 & m = m_\star \\ \min_{r \in Ch_{\tilde{\mathcal{G}}'}(m)} \eta(r; \tilde{\mathcal{G}}', f, b, h) & \text{all other } m \end{cases} \quad (37)$$

$$\eta(r; \tilde{\mathcal{G}}', f, b, h) = \begin{cases} \infty & \psi(r; \tilde{\mathcal{G}}', f, b, h) = 0 \\ \sum_{m' \in Ch_{\tilde{\mathcal{G}}'}(r)} \eta(m'; \tilde{\mathcal{G}}', f, b, h) + \log \frac{\psi(m'; \mathcal{G}', f, b, h)}{\psi(r; \mathcal{G}', f, b, h)} & \psi(r; \mathcal{G}', f, b, h) < \infty \,, \end{cases} \quad (38)$$

where the sum in equation 38 was introduced because reactions in $\mathcal{G}'$ have only one parent. These equations correspond precisely to minimum cost equations from AO*, allowing algorithms from Chakrabarti (1994) to be applied. $\rho$ can then be straightforwardly recovered from $\eta$.

However, since the sum in equation 38 has only one element, this is in fact equivalent to path-finding in an ordinary graph, so Dijkstra's algorithm could be used to solve for $\eta$ in log-linear time (once $\psi$ is solved). □

# E    DISCUSSION OF PREVIOUS SEARCH ALGORITHMS AND WHY THEY MIGHT STRUGGLE TO OPTIMIZE SSP

This section aims to provide a more detailed discussion of previously-proposed algorithms. It is structured into 3 parts. In E.1, we qualify the content of this section by explaining how the performance of algorithms will depend on a lot of factors, and therefore we cannot really say that any algorithm will be incapable of maximizing SSP. In E.2 we review previous algorithms and state how they might be configured to maximize SSP. For most algorithms, the closest configuration to maximizing SSP is to reward individual synthesis plans with high success probability. Section E.3 provides an argument of why this may not always maximize SSP.

## E.1    CAN WE SAY ANYTHING DEFINITIVE ABOUT THE ABILITY OF DIFFERENT ALGORITHMS TO MAXIMIZE SSP?

The behaviour of a search algorithm can depend on many configurable parameters (including function-valued parameters):

1. Reward functions (or similar): algorithms will behave differently based on the kinds of synthesis plans which are rewarded.
2. Heuristics: most heuristic-guided search algorithms will, to some degree, follow the guidance of the search heuristic.
3. Other hyperparameters (e.g. the exploration constant in MCTS).

How can these parameters be adjusted? One option is to set these parameters independently of $\xi_f$ or $\xi_b$ (i.e. completely ignoring them). Even under these conditions, an algorithm could possibly return a set of synthesis plans with high SSP. This could happen due to random chance (e.g. lucky choice of nodes to expand), or due to some kind of "alignment" between the algorithm's internals, $\xi_f$, and $\xi_b$ (for example, reactions with high feasibility having low cost). However, such outcomes are clearly not attainable *systematically* without accessing $\xi_f$ and $\xi_b$, since random chance is not repeatable, and "alignment" for one feasibility/buyability model necessarily means misalignment for another one. Therefore, we argue it only makes sense to compare algorithms' ability to maximize SSP when they are configured to use information from $\xi_f$ and $\xi_b$ to maximize SSP.

Even under these conditions however, given a particular $m_\star$, it is likely possible to design a custom heuristic and setting of the algorithm's parameters which will lead an algorithm to maximize SSP very effectively for that particular $m_\star$. This makes it difficult [perhaps impossible] to prove statements like "algorithm A is fundamentally incapable of effectively maximizing SSP." At the same time, given any configured version of an algorithm (including a search heuristic, reward, etc.), it is likely possible to find a particular $m_\star$ and feasibility model $\xi_f$, $\xi_b$ where the algorithm fails catastrophically (a little bit like the no free lunch theorem in machine learning). This makes it difficult [perhaps impossible] to prove statements like "algorithm A is generally better than algorithm B at maximizing SSP."

Therefore, when discussing previous algorithms in this section, we merely adopt the goal of showing how the algorithm cannot be configured to directly and straightforwardly maximize SSP in the same manner as retro-fallback.

## E.2    EXISTING ALGORITHMS AND THEIR CONFIGURABILITY

### E.2.1    BREADTH-FIRST SEARCH

**Description of algorithm**    A very basic search algorithm: expand frontier nodes in the order they were added.

**Configurable inputs/parameters**    None.

**How to configure to maximize SSP?**    N/A

**Potential modifications to the algorithm that could help?**    We see no obvious changes.

### E.2.2 MONTE-CARLO TREE SEARCH (MCTS)

**Description of algorithm** Used in (Segler et al., 2018; Coley et al., 2019b; Genheden et al., 2020). MCTS creates an MDP where "states" are set of molecules and "actions" are reactions which react one molecule in a state and replace it with other molecules. This corresponds to the "OR" tree introduced in section B.3. At each step, it descends the tree choosing nodes which maximize

$$\frac{W(s_t, a)}{N(s_t, a)} + cP(s_t, a)\frac{\sqrt{N(s_{t-1}, a_{t-1})}}{1 + N(s_t, a)} \, , \tag{39}$$

where $W(s_t, a)$ is the total reward accumulated while taking action $a$ to reach state $s_t$, $N(s_t, a)$ is the number of times when the algorithm has performed action $a$ to reach state $s_t$ and $P(s_t, a)$ is some sort of prior probability of performing action $a$ to reach $s_t$, and $c$ is a constant. The algorithm is designed to eventually converge to the action sequence which maximizes reward.

**Configurable inputs/parameters** The reward function (mapping states to scalar rewards) $R$, the search heuristic $V$ (an estimate of the value function, e.g. based on rollouts), the policy / prior $P$, and the UCB exploration constant $c$.

**How to configure to maximize SSP?** Rewards and value functions in MCTS depend on *individual* synthesis plans, so a reward of $\sigma(T; \xi_b; \xi_f)$ would reward individually successful synthesis plans. In practice, this reward could be estimated with samples.

**Potential modifications to the algorithm that could help?** One option is to make the reward and policy change over time. For example, one could have the reward be the *additional* SSP gained from discovering a new plan. However, it is possible that MCTS will not behave well in this scenario: the principle of MCTS is to narrow down on the best sequence of actions by slowly tightening a confidence interval around their expected return. However, if the rewards change over time then these interval estimates will likely become inaccurate. Although the intervals could be widened (e.g. by increasing $c$) this will result in MCTS behaving more randomly and not searching as efficiently. Also, we note that there are many possible design choices here, and further probing of alternative options might yield improved results.

### E.2.3 DEPTH-FIRST PROOF NUMBER SEARCH (WITH HEURISTIC EDGE INITIALIZATION)

**Description of algorithm** Proposed in Heifets & Jurisica (2012) and augmented by Kishimoto et al. (2019). Basic proof number search assigns "proof numbers" and "disproof numbers" which represent the number of nodes needed to prove or disprove whether there exist a synthesis plan to synthesize a given molecule and selects nodes using this information. The heuristic edge initialization Kishimoto et al. (2019) uses a heuristic to initialize the proof and disproof numbers for each reaction.

**Configurable inputs/parameters** The edge initialization heuristic $h$ and threshold functions to control depth-first vs breadth-first search behaviour.

**How to configure to maximize SSP?** This algorithm inherently takes a very binary view of retrosynthesis (seeing nodes as either proven or disproven), and therefore is not very amenable to maximizing SSP. At best one could change the heuristic values to reflect feasibilities.

**Potential modifications to the algorithm that could help?** Aside from changing the definitions of proof/disproof numbers, we do not see any options here.

### E.2.4 RETRO*

**Description of algorithm** An algorithm for minimum cost search in AND/OR trees (Chen et al., 2020), essentially identical to the established AO* algorithm (Chang & Slagle, 1971; Martelli & Montanari, 1978; Nilsson, 1982; Mahanti & Bagchi, 1985). At each step, the algorithm selects a potential synthesis plan $T$ with minimal estimated cost, where the cost of a synthesis plan is defined

as

$$c_{\text{total}}(T) = \sum_{r \in T} c_R(r) + \sum_{m \in T} c_M(m)$$

(i.e. a sum of individual costs for each reaction and molecule), then expands one frontier node from $T$. A heuristic is used to set the costs $c_M$ for frontier molecules.

**Configurable inputs/parameters** The parameters of retro* are the costs of each molecule and reaction, and the heuristic functions.

**How to configure to maximize SSP?** SSP does not [in general] decompose into a straightforward sum or product of values for every node. Most likely, the closest proxy to SSP would be to set a cost for each molecule and reaction with $\xi_f$ and $\xi_b$. One option is to set the cost of each reaction/molecule to the negative log of its marginal feasibility/buyability:

$$c_M(m) = -\log \mathbb{E}_{b \sim \xi_b} [b(m)]$$
$$c_R(r) = -\log \mathbb{E}_{f \sim \xi_f} [f(r)].$$

This option does have a somewhat principled interpretation: if all feasibility/buyability outcomes are independent then the cost of a plan is just the negative log of its joint SSP. Of course, this relationship does not hold in the general non-independent case. However, we do not see a way to adjust this formula to account for non-independence in general, so we suggest using this setting in all cases.

**Potential modifications to the algorithm that could help?** One could potentially change the reaction and molecule costs with time to account for changes elsewhere in the search graph. For example, reactions/molecules which are already part of a feasible plan could be re-assigned higher costs to make the algorithm search for non-overlapping plans. However, this strategy seems unlikely to work in general: for some search graphs it is possible that the best backup plans will share some reactions with established synthesis plans. We were unable to come up with a strategy that did not have obvious and foreseeable failure modes, so we decided not to pursue this direction.

### E.2.5 RETROGRAPH

**Description of algorithm** Proposed in Xie et al. (2022), this algorithm functions like a modified version of retro* but on a minimal AND/OR graph instead of a tree. A graph neural network is used to prioritize nodes.

**Configurable inputs/parameters** Like retro*, this method uses additive costs across reactions and a heuristic functions, but also uses a graph neural network to select promising nodes.

**How to configure to maximize SSP?** Similar to retro*, we believe the closest proxy is to set costs equal to the negative log marginal feasibility/buyability and minimize this.

**Potential modifications to the algorithm that could help?** We could not think of anything.

### E.3 FINDING INDIVIDUALLY SUCCESSFUL SYNTHESIS PLANS MAY NOT BE ENOUGH TO MAXIMIZE SSP

Since the best configuration for most existing algorithms to maximize SSP appears to be to reward finding individual synthesis plans with high success probability, here we examine to what degree this objective overlaps with optimizing SSP. Clearly the objectives are *somewhat* related: SSP will always be at least as high as the success probability of any individual synthesis plan considered. However, it is not difficult to imagine cases where these objectives diverge.

Figure E.1 illustrates such a case, wherein a synthesis plan with reactions $r_1, r_2$ has been found and the algorithm must choose between expanding $m_3$ or $m_4$. When considering whether to expand $m_3$ however, it must be noted that any new synthesis route proceeding via $r_3$ will also use $r_1$, and therefore will only increase SSP for samples $f, b$ where $f(r_1) = 1$ *and* $f(r_3) = 1$ *and* $f(r_2) = 0$ (or $b(m_2) = 0$). For example, if $\mathbb{P}_f[f(r_2) = 0]$ is very small, or the feasibility of $r_2$ and $r_3$ are highly

Figure E.1: AND/OR graph illustrating how maximizing SSP can be different from finding individually successful synthesis plans (cf. E.3). Green nodes are part of an existing synthesis plan with non-zero success probability, red nodes are not.

correlated, then it is very unlikely that expanding $r_3$ will increase SSP, even if the $\{r_1, r_3\}$ synthesis plan may have an individually high success probability. In these cases expanding $m_4$ would be a better choice. In other cases, expanding $m_3$ would be the better choice.

This exactly illustrates that to maximize SSP *beyond* finding an individually successful synthesis plan, an algorithm would clearly need to account for the statistical dependencies between existing and prospective synthesis plans in its decision-making, which simply is not possible by reasoning about individual synthesis plans in isolation. This provides compelling motivation to develop algorithms which can reason directly about sets of synthesis plans.

# F    EXTENDED RELATED WORK (EXCLUDING PREVIOUS SEARCH ALGORITHMS)

Here we cite and discuss papers which are relevant to retrosynthesis, but do not propose multi-step search algorithms (which are discussed in Appendix E).

## F.1    SEARCH HEURISTICS FOR RETROSYNTHESIS

Many papers propose search heuristics for retrosynthesis algorithms, including rollouts (Segler et al., 2018), parametric models (Chen et al., 2020), and a variety of heuristics informed by chemical knowledge (Schwaller et al., 2020; Ertl & Schuffenhauer, 2009; Thakkar et al., 2021; Li & Chen, 2022). Many papers also propose to learn heuristics using techniques from machine or reinforcement learning, where a heuristic is learned based on previous searches or data (Coley et al., 2018; Liu et al., 2022; Kim et al., 2021; Yu et al., 2022; Liu et al., 2023a). A potential point of confusion is that some of these works describe their contribution as a "retrosynthesis algorithm" or "retrosynthetic planning algorithm." Given that the end product of these papers is a value function (or cost estimate) which is plugged into a previously proposed algorithm (typically MCTS or retro*), we think these papers should be more accurately viewed as proposing *heuristics*. The heuristic is orthogonal to the underlying search algorithm, so we view these works as complementary rather than competitive. We hope in the future to investigate learning heuristics for retro-fallback using similar principles.

## F.2    GENERATIVE MODELS

Several works propose parametric generative models of synthesis plans (Bradshaw et al., 2019; 2020; Gottipati et al., 2020; Gao et al., 2021). Although this resembles the goal of explicit search algorithms, such generative models are fundamentally limited by their parametrization: they have no guarantee to find a synthesis plan if it exists, and are often observed to fail to produce a valid synthesis plan in practice (Gao et al., 2021). We think such models are best viewed as trying to *amortize* the output of an explicit planning algorithms, making them more similar in spirit to search heuristics (F.1).

## F.3    SINGLE-STEP RETROSYNTHESIS

Many models have been proposed to predict possible chemical reactions, including template classifiers (Segler & Waller, 2017; Seidl et al., 2021), graph editing methods (Dai et al., 2019; Sacha et al., 2021; Chen & Jung, 2021), and transformers (Irwin et al., 2022; Zhong et al., 2022; Liu et al., 2023b). Such models are a useful component of a retrosynthesis algorithm, but do not themselves perform *multi-step* retrosynthesis.

## F.4    FUSIONRETRO

One work which does not fit nicely into any of the previous subsections is FusionRetro (Liu et al., 2023b). On one level, the paper describes a reaction prediction model based on a transformer, which is essentially a single-step reaction prediction model (F.3). However, unlike other models which just condition on a single input molecule, in FusionRetro the predictions are conditioned on all predecessor molecules in a multi-step search graph. The paper describes an inference procedure to make predictions from the model autoregressively, which resembles both a generative model for synthesis plans (F.2) or a pruning heuristic for breadth-first search (F.1). A significant portion of the paper also describes benchmarking and evaluation of plan quality.

We think that FusionRetro and retro-fallback can both be viewed as responses to unrealistically lenient evaluation metrics used in prior works on retrosynthesis (chiefly reporting success if a "solution" is found without any regard to whether the solution is realistic). Liu et al. (2023b)'s general response is to evaluate the quality of entire plans rather than individual steps, and perform this evaluation using entire synthesis plans from the literature. The advantage of this approach is that it is close to ground-truth data, but has the disadvantage that high-quality ground truth data is fairly scarce, especially for long plans involving rare reactions. In contrast, our response is to model uncertainty about reactions and use this uncertainty in evaluation (to define SSP). The advantage of our

approach is that it does not [necessarily] require any data, while the disadvantage is that it requires a good model of reaction uncertainty, which we currently do not have (and creating such a model is likely to be difficult).

Critically, the approaches described in these papers are *not* mutually exclusive: a backward reaction model which depends on the entire search graph $\mathcal{G}'$ (such as FusionRetro) could be used in retro-fallback, while the quality of synthesis plans proposed by a method like FusionRetro could be evaluated using SSP. We leave combining and building upon these works in more realistic retrosynthesis programs to future work.

### F.5    PLANNING IN STOCHASTIC GRAPHS

Prior works have also considered planning in stochastic graphs, albeit in other contexts. For example, the "Canadian Traveller Problem" and its variants (Papadimitriou & Yannakakis, 1991) study search on a graph where edges can be randomly deleted.[12]   However, this is an *online* problem, meaning that the planning algorithm learns about edge deletions during the planning process.  In contrast, our algorithm assumes *offline* planning because chemists desire complete synthesis plans *before* performing any lab experiments.  Moreover, works in this area seem to assume explicit OR graphs with independent probabilities, while our problem uses implicit AND/OR graphs with non-independent probabilities.

---

[12]The original problem statement was to find a path between two cities in Canada, where roads may be randomly blocked by snow and force the driver to turn back and find an alternative path (in reality however, Canadians simply drive through the snow).

## G    EXTENDED EXPERIMENT SECTION

### G.1    DETAILS OF EXPERIMENTAL SETUP

#### G.1.1    SOFTWARE IMPLEMENTATION

Our code is available at `https://github.com/AustinT/retro-fallback-iclr24`.
We built our code around the open-source library SYNTHESEUS[13] (Maziarz et al., 2023) and used its
implementations of retro* and MCTS in our experiments. The exact template classifier from Chen
et al. (2020) was used by copying their code and using their model weights. Our code benefitted
from the following libraries:

- `pytorch` (Paszke et al., 2019), `rdkit`[14] and `rdchiral` (Coley et al., 2019a). Used in
  the template classifier.

- `networkx` (Hagberg et al., 2008). Used to store search graphs and for analysis.

- `numpy` (Harris et al., 2020), `scipy` (Virtanen et al., 2020), and `scikit-learn` (Pe-
  dregosa et al., 2011). Used for array programming and linear algebra (e.g. in the feasibility
  models).

#### G.1.2    FEASIBILITY MODELS

As stated in section 6, we examined four feasibility models for this work, which assign different
marginal feasibility values and different correlations between feasibility outcomes. The starting
point for our feasibility models was the opinion of a trained organic chemist that around 25% of the
reactions outputted by the pre-trained template classification model from Chen et al. (2020) were
"obviously wrong". From this, we proposed the following two marginal values for feasibility:

1. (C) A constant value of $1/2$ for all reactions. This is an attempt to account for the 25% of
   reactions which were "obviously wrong", plus an additional unknown fraction of reactions
   which seemed plausible but may not work in practice. Ultimately anything in the interval
   $[0.2, 0.6]$ seemed sensible to use, and we chose $1/2$ as a nice number.

2. (R) Based on previous work with template classifiers suggesting that the quality of the
   proposed reaction decreases with the softmax value (Segler & Waller, 2017; Segler et al.,
   2018), we decided to assign higher feasibility values to reactions with high softmax values.
   To avoid overly high or low feasibility values, we decided to assign values based on the *rank*
   of the outputted reaction, designed the following function which outputs a high feasibility
   ($\approx 75\%$) for the top reaction and decreases to ($\approx 10\%$) for lower-ranked reactions:

$$p(\text{rank}) = \frac{0.75}{\text{rank}/10} . \tag{40}$$

Note that "rank" in the above equation starts from 1.

We then added correlations on top of these marginal feasibility values. The independent model is
simple: reaction outcomes are sampled independently using the marginal feasibility values described
above. To introduce some correlations without changing the marginal probabilities, we created the
following probabilistic model which assigns feasibility outcomes by applying a threshold to the
value of a latent Gaussian process (Williams & Rasmussen, 2006):

$$\text{outcome}(z) = \mathbf{1}_{z>0} \tag{41}$$
$$z(r) \sim \mathcal{GP}\left(\mu(\cdot), K(\cdot, \cdot)\right) \tag{42}$$
$$\mu(r) = \Phi^{-1}\left(\mathbb{P}[f(r) = 1]\right) \tag{43}$$
$$K(r, r) = 1 \quad \forall r \tag{44}$$

Here, $\Phi$ represents the CDF of the standard normal distribution. Because of equation 44, the
marginal distribution of each reaction's $z$ value is $\mathcal{N}(\Phi^{-1}(p(r)), 1)$ which will be positive with

---

[13]https://github.com/microsoft/syntheseus/

[14]Specifically version 2022.09.4 (Landrum et al., 2023).

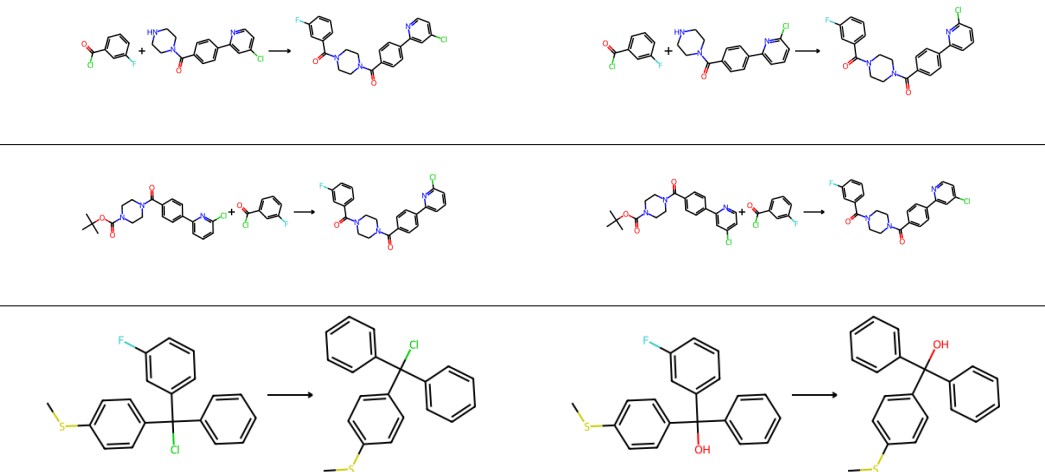

Figure G.1: Sample pairs of reactions where $K_{\text{total}} > 0.8$. **Top:** both reactions join a COCl group to an NH group in a ring to form molecules which differ only by the location of the Cl atom on the right side ring (far away from the reaction site). **Middle:** two reactions transforming a tert-butyl ester into a ketone with a fluorine-containing ring (difference between reactions is the location of the Cl atom on the ring far away from the reaction site). **Bottom:** two reactions removing a fluorine atom from an aromatic ring on similar molecules (difference is between the Cl and OH groups). **Summary:** these pairs of reactions are all very similar.

probability $\mathbb{P}[f(r) = 1]$ (i.e. it preserves arbitrary marginal distributions). If $K$ is the identity kernel (i.e. $K(r_1, r_2) = \mathbf{1}_{r_1 = r_2}$) then this model implies all outcomes are independent. However, non-zero off-diagonal values of $K$ will induce correlations (positive or negative).

We aimed to design a model which assigns correlations very conservatively: only reactions involving similar molecules *and* which induce similar changes in the reactant molecules will be given a high positive correlation; all other correlations will be near zero. We therefore chose a kernel as a product of two simpler kernels:

$$K_{\text{total}}(r_1, r_2) = K_{\text{mol}}(r_1, r_2) K_{\text{mech}}(r_1, r_2) \ .$$

We chose $K_{\text{mol}}(r_1, r_2)$ to be the Jaccard kernel

$$k(x, x') = \frac{\sum_i \min(x_i, x'_i)}{\sum_i \min(x_i, x'_i)}$$

between the Morgan fingerprints (Rogers & Hahn, 2010) with radius 1 of the entire set of product and reactant molecules.[15] We chose $K_{\text{mech}}(r_1, r_2)$ to be the Jaccard kernel of the absolute value of the *difference* between the product and reactant fingerprints individually. The difference vector between two molecular fingerprints will essentially yield the set of subgraphs which are added/removed as part of the reaction. For this reason, it has been used to create representations of chemical reactions in previous work (Schneider et al., 2015).

We illustrate some outputs of this kernel in Figures G.1–G.3. Figure G.1 shows that reactions with a high kernel value ($> 0.8$) are generally quite similar, both in product and in mechanism. Figure G.2 shows that reactions with modest similarity values in $[0.4, 0.6]$ have some similarities but are clearly less related. Figure G.3 shows that reactions with low similarity values in $[0.05, 0.1]$ are generally quite different. After a modest amount of exploratory analysis we were satisfied that this kernel behaved as we intended, and therefore used it in our experiments without considering further alternatives. However, we imagine there is room for improvement of the kernel in future work to better align with the beliefs of chemists.

For feasibility models based on Gaussian processes, drawing $k$ independent samples for a set of $N$ reactions will generally scale as $\mathcal{O}(kN^3)$. This makes the feasibility model expensive for longer

---

[15]This is the same as adding the fingerprint vectors for all component molecules.

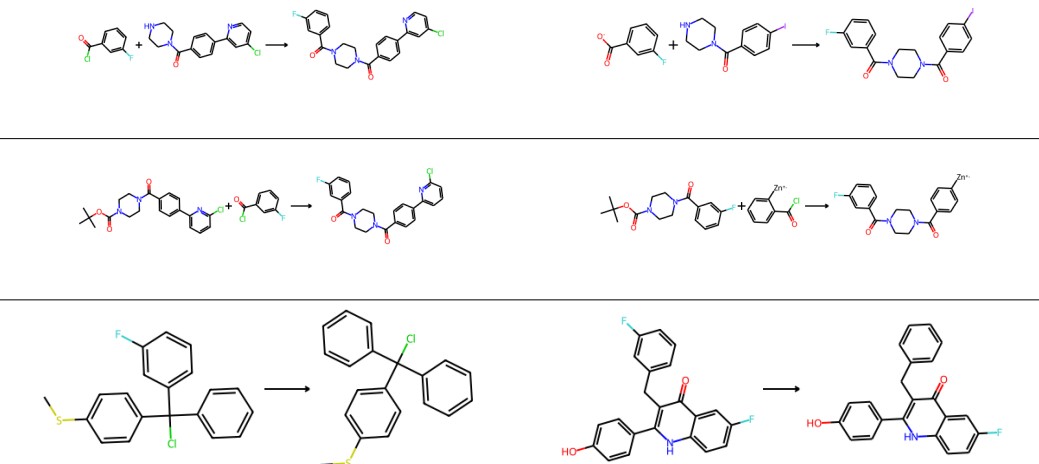

Figure G.2: Examples of reactions where $0.4 \leq K_{\text{total}} \leq 0.6$. **Top:** similar conjugation reactions, but the reactant on the right side is now a $COO^-$ anion instead of a COCl group. **Middle:** similar reaction, although on the right reaction has a $Zn^+$ on the ring instead of F. **Bottom:** two reactions which remove a fluorine atom from an aromatic ring but on molecules which are much less similar than Figure G.1. **Summary:** these pairs of reactions have similarities but are less similar than the reactions in Figure G.1.

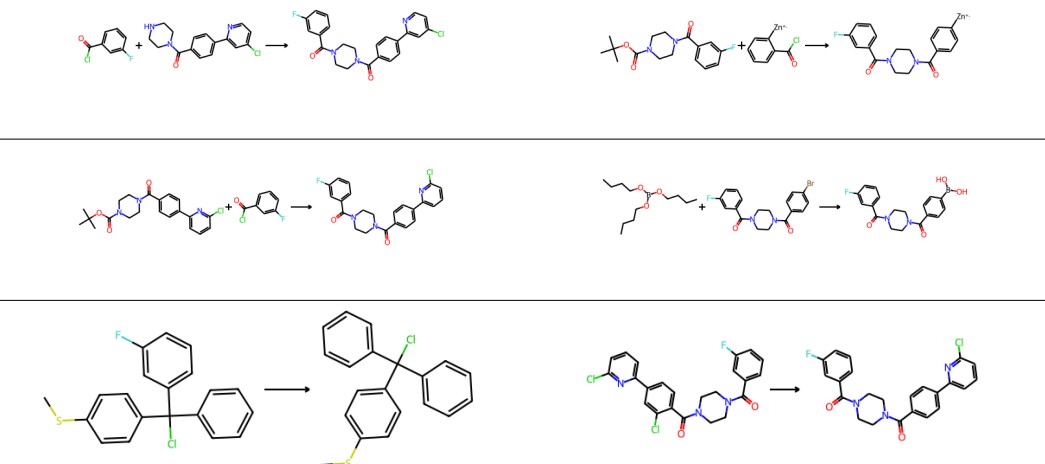

Figure G.3: Examples of reactions where $0.05 \leq K_{\text{total}} \leq 0.1$. The pairs of reactions are generally quite different.

searches. Other feasibility models which induce correlations are likely to have similar scaling. However, for this particular kernel, approximate samples could be drawn in $\mathcal{O}(kN)$ time by using a random features approximation for the Jaccard kernel (Tripp et al., 2023).

### G.1.3 BUYABILITY MODELS

Following Chen et al. (2020) we based our buyability models on the inventory of eMolecules: a chemical supplier which acts as a middleman between more specialized suppliers and consumers. According to eMolecule's promotional material[16], they offer 6 "tiers" of molecules:

0. (*Accelerated Tier*). "Delivered in 2 days or less, guaranteed. Most reliable delivery service. Compound price is inclusive of a small service fee, credited back if not delivered on time. Available in the US only."

1. "Shipped within 1- 5 business days. Compounds from suppliers proven to ship from their location in $< 5$ days."

2. "Shipped within 10 business days. Compounds from suppliers proven to ship from across the globe in $< 10$ days"

3. "Shipped within 4 weeks. Shipped from suppliers further from your site and often with more complex logistics. Synthesis may be required using proven reactions."

4. "Shipped within 12 weeks. Usually requires custom synthesis on demand."

5. "Varied ship times. Requires custom synthesis for which a quote can be provided on request."

Much like machine learning researchers, chemists usually want to complete experiments as quickly as possible and probably would prefer not to wait 12 weeks for a rare molecule to be shipped to them. Such molecules could arguably be considered less "buyable" on this subjective basis alone, so we decided to create buyability models based on the tier of molecule. Unfortunately, the public repository for retro* does not contain any information on the tier of each molecule, and because their inventory was downloaded in 2019 this information is no longer available on eMolecules' website. Therefore we decided to re-make the inventory using the latest data.

We downloaded data from eMolecules downloads page[17], specifically their "orderable" molecules and "building blocks" with quotes. After filtering out a small number of molecules (31407) whose SMILES were not correctly parsed by rdkit we were left with 14903392 molecules with their associated purchase tiers. Based on this we created 2 buyability models:

- **Binary:** all molecules in tiers 0-2 are purchasable with 100% probability. Corresponds to realistic scenario where chemists want to do a synthesis and promptly.

- **Stochastic:** molecules are independently purchasable with probability that depends on the tier (100% for tiers 0-2, 50% for tier 3, 20% for tier 4, 5% for tier 5). These numbers were chosen as subjective probabilities that the compounds would be delivered within just 2 weeks (shorter than the longer times advertised). This still corresponds to a chemist wanting to do the synthesis within 2 weeks, but being willing to risk ordering a molecule whose stated delivery time is longer.

All of the experiments in this text (except for G.2.4) were run using the *binary* buyability model. In the future, we believe that better buyability models could be formed by introducing correlations between molecules coming from the same supplier, but we do not investigate that here (chiefly because the eMolecules data we downloaded does not contain information about suppliers).

### G.1.4 TEST MOLECULES

The 190 test molecules were accessed using the `syntheseus` wrapper package for this benchmark.[18]

To include molecules with a wider range of synthetic difficulties, we also performed experiments on a set of 1000 randomly selected molecules from the GuacaMol test set (Brown et al., 2019). These test molecules were generated with the following procedure:

---

[16]At the time of publication, this was available at: `https://21266482.fs1.hubspotusercontent-na1.net/hubfs/21266482/GUCHBBXX-E-02.01-0322_eMolecules%20Tier%20Guide.pdf`

[17]Downloaded 2023-09-08.

[18]Available at: `https://pypi.org/project/syntheseus-retro-star-benchmark/`

1. Download the publicly available test set from Brown et al. (2019)

2. Filter our all molecules available in the eMolecules inventory (G.1.3)

3. Shuffle all molecules and take the first 1000

Code to reproduce this process, and the entire test set in shuffled order is included in our code.

Finally, we also ran an experiment with the test data from FusionRetro (Liu et al., 2023b), using the test dataset found on their GitHub repository.

### G.1.5   ALGORITHM CONFIGURATION

Retro-fallback was run with $k = 256$ samples from $\xi_f, \xi_b$. A graph-structured AND/OR search graph was used (which may contain cycles). $s(\cdot)$, $\psi(\cdot)$, and $\rho(\cdot)$ were solved by iterating the recursive equations (including around cycles) until convergence (if this did not occur we reset all values to 0 and resumed iteration).

All other algorithms were configured to maximize SSP, as described in Appendix E.2. In particular, this means:

- Breadth-first search was run with no modifications, using the implementation from `syntheseus`.

- retro* was run using $-\log \mathbb{E}_f[f(r)]$ as the reaction cost and $-\log \mathbb{E}_b[b(m)]$

- MCTS was run using $\sigma(T; \xi_f, \xi_b)$ as the reward for finding synthesis plan $T$ (empirically estimated from a finite number of samples). To allow the algorithm to best make use of its budget of reaction model calls, we only expanded nodes after they were visited 10 times. The marginal feasibility value of reach reaction was used as the policy in the upper-confidence bound. We used an exploration constant of $c = 0.01$ to avoid "wasting" reaction model calls on exploration, and only gave non-zero rewards for up to 100 visits to the same synthesis plan to avoid endlessly re-visiting the same solutions.

All of these algorithms are run with a *tree*-structured search graph (which for MCTS is an "OR" graph; AND/OR for all others).

We chose *not* to compare with proof-number search (Kishimoto et al., 2019) because we did not see a way to configure it to optimize SSP (see Appendix E.2.3). We chose not to compare with algorithms requiring some degree of learning from self-play (including RetroGraph and the methods discussed in Appendix F.1) due to computational constraints, and because it seemed inappropriate to compare with self-play methods without also learning a heuristic for retro-fallback with self-play.

### G.1.6   HEURISTIC FUNCTIONS

The heuristic obviously plays a critical role in heuristic-guided search algorithms! Ideally one would control for the effect of the heuristic by using the same heuristic for different methods. However, this is not possible when comparing algorithms from different families because the heuristics are interpreted differently! For example, in retro-fallback the heuristic is interpreted as a potential future SSP value in $[0, 1]$ (higher is better), while in retro* it is interpreted as a cost between $[0, \infty]$ (lower is better). If we used literally the same heuristic it would give opposite signals to both of these algorithms, which is clearly not desirable or meaningful. Therefore, we tried our best to design heuristics which were "as similar as possible."

**Optimistic heuristic**   Heuristics which predict the best possible value are a common choice of naive heuristic. Besides being an important baseline, optimistic heuristics are always *admissible* (i.e. they never overestimate search difficulty), which is a requirement for some algorithms like A* to converge to the optimal solution (Pearl, 1984). For retro-fallback, the most optimistic heuristic is $h_{\mathrm{rfb}}(m) = 1$, while for retro* it is $h_{\mathrm{r*}}(m) = 0$, as these represent the best possible values for SSP and cost respectively. For MCTS, the heuristic is a function of a *partial plan* $T'$ rather than a single molecule. We choose the heuristic to be $\mathbb{E}_{f \sim \xi_f}[\min_{r \in T'} f(r)]$, which is the expected SSP of the

plan $T'$ if it were completed by making every frontier molecule buyable.[19] In practice this quantity was estimated from $k$ samples (same as retro-fallback).

**SA score heuristic**   SA score gives a molecule a score between 1 and 10 based on a dictionary assigning synthetic difficulties to different subgraphs of a molecule (Ertl & Schuffenhauer, 2009). A score of 1 means easy to synthesize, while a score of 10 means difficult to synthesize. For retro-fallback, we let the heuristic decrease linearly with the SA score:

$$h_{\mathrm{rfb}}(m) = 1 - \frac{\mathrm{SA}(m) - 1}{10} \ .$$

Because the reaction costs in retro* were set to negative log feasibility values, we thought a natural extension to retro* would be to use

$$h_{\mathrm{r*}}(m) = -\log h_{\mathrm{rfb}}(m) \ .$$

This choice has the advantage of preserving the interpretation of total cost as the negative log joint probability, which also perfectly matches retro-fallback's interpretation of the heuristic (recall that in section 4.1 the heuristic values were assumed to be independent). We designed MCTS's heuristic to also match the interpretation of "joint probability":

$$h_{\mathrm{MCTS}}(T') = \mathbb{E}_{f \sim \xi_f} \left[ \left( \underbrace{\min_{r \in T'} f(r)}_{\text{reactions feasible}} \right) \prod_{m \in \mathcal{F}(T'), b(m)=0} h_{\mathrm{rfb}}(m) \right]$$

which is the expected SSP of the plan if all non-purchasable molecules are made purchasable independently with probability $h_{\mathrm{rfb}}(m)$.

### G.1.7   COMPUTING ANALYSIS METRICS

Our primary analysis metric is the SSP. For algorithms which use AND/OR graphs (retro-fallback, retro*, breadth-first search), we computed the SSP using equations 4–5 with $k = 10\,000$ samples from $\xi_f, \xi_b$. For algorithms using a tree-structured search graph, this was pre-processed into a cyclic search graph before analysis (for consistency).

For algorithms which use OR trees (in this paper, just MCTS) the best method for analysis is somewhat ambiguous. One option is to extract all plans $T \subseteq \mathcal{G}'$ and calculate whether each plan succeeds on a series of samples $f_i, b_i$. A second option is to convert $\mathcal{G}'$ into an AND/OR graph and analyze it like other AND/OR graphs. Although they seem similar, these options are subtly different: an OR graph may contain reactions in different locations which are not connected to form a synthesis plan, but *could* form a synthesis plan if connected. The process of converting into an AND/OR graph would effectively form all possible synthesis plans which could be made using reactions in the original graph, even if they are not actually present in the original graph. We did implement both methods and found that converting to an AND/OR graph tends to increase performance, so this choice does make a meaningful difference. We think the most "realistic" option is unclear, so for consistency with other algorithms we chose to just convert to an AND/OR graph.

All analysis metrics involving individual synthesis routes were calculated by enumerating the routes in best-first order using a priority queue, implemented in SYNTHESEUS.

---

[19]Note that the $\min$ function will be 1 if *all* reactions are feasible, otherwise 0. Using $\prod_r$ instead of $\min_r$ would yield the same output.

## G.2    ADDITIONAL RESULTS FOR SECTION 6.2 (HOW EFFECTIVE IS RETRO-FALLBACK AT MAXIMIZING SSP)

Figure G.4 shows that if the optimistic heuristic is used instead of the SA score heuristic, retro-fallback still maximizes SSP more effectively than other algorithms.

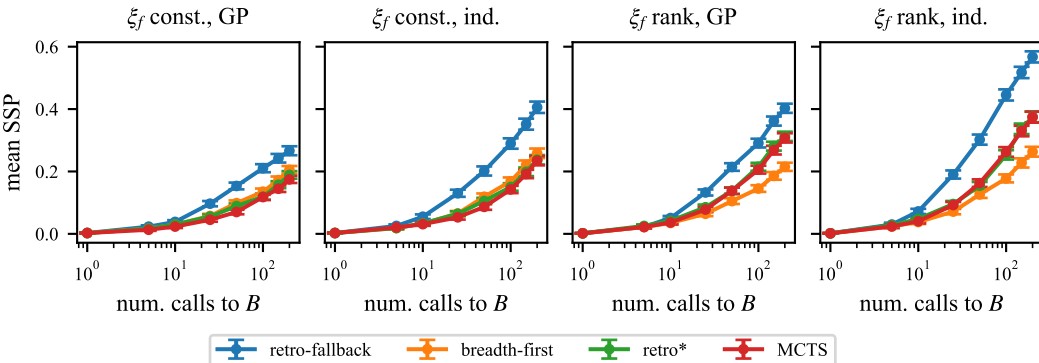

Figure G.4:    Mean SSP across all 190 test molecules from Chen et al. (2020) vs time using the *optimistic* heuristic. Interpretation is identical to Figure 2 (except for the different heuristic).

Figure G.5 shows a similar result for the easier GuacaMol test molecules. However, the difference between retro-fallback and other algorithms is smaller than on the 190 "hard" molecule test set from Chen et al. (2020). This is likely because this test set contains many molecules which are easier to synthesize. For example, after a single iteration algorithms achieve a mean SSP of $\approx 0.15$ on the GuacaMol test set, compared to $0$ for the molecules from Chen et al. (2020).

We also note the following:

- For the "constant, independent" feasibility model, breadth-first search and retro* are essentially the same algorithm. Therefore their performance is almost identical.
- Comparing results with the SA score and optimistic heuristic, it appears that the use of a non-trivial heuristic does not actually make a terribly large difference. This is consistent with the results of Chen et al. (2020) in their experiments with retro*, where adding a heuristic provided only a modest increase in performance.

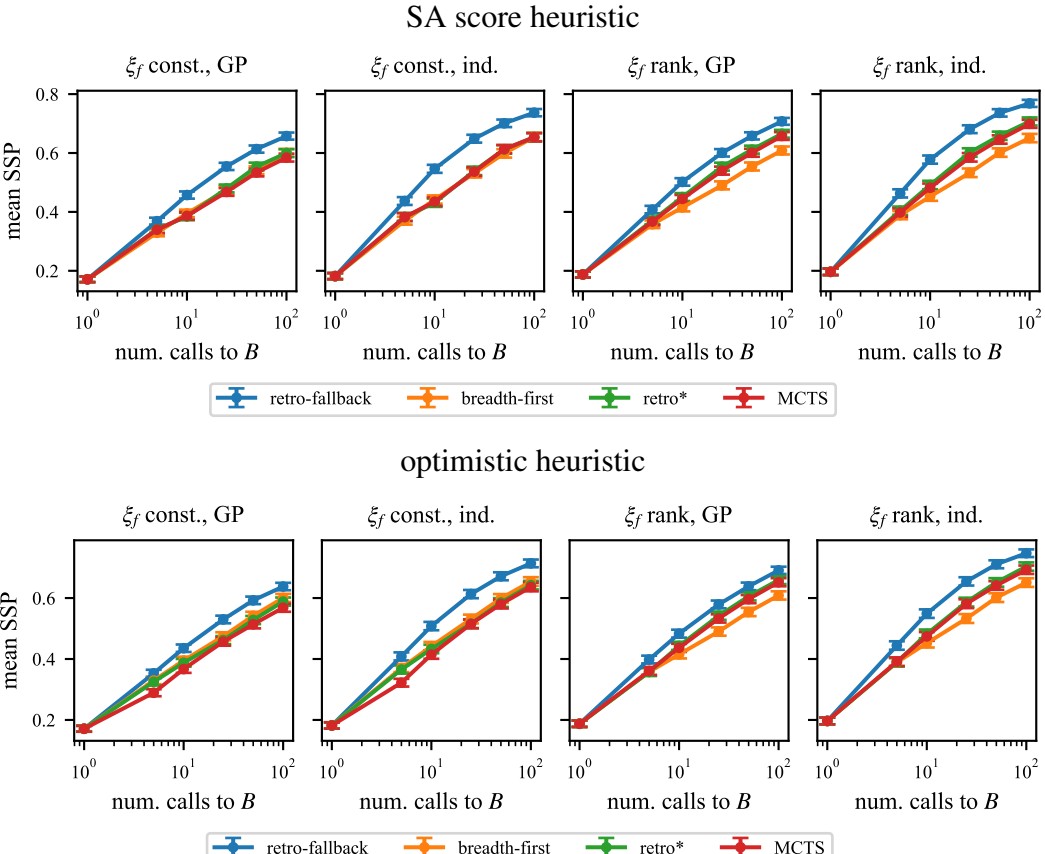

Figure G.5: Mean SSP vs time for 1000 molecules from GuacaMol (described in G.1.4). Interpretation is identical to Figure 2.

### G.2.1 RESULTS FOR INDIVIDUAL MOLECULES

The SSP results in Figures 2, G.4, and G.5 are aggregated across all test molecules. To understand where performance differences come from, Figures G.6 and G.7 plot SSP over time for individual molecules. It appears that there is a high amount of variation due to randomness, and variation in outcomes between molecules.

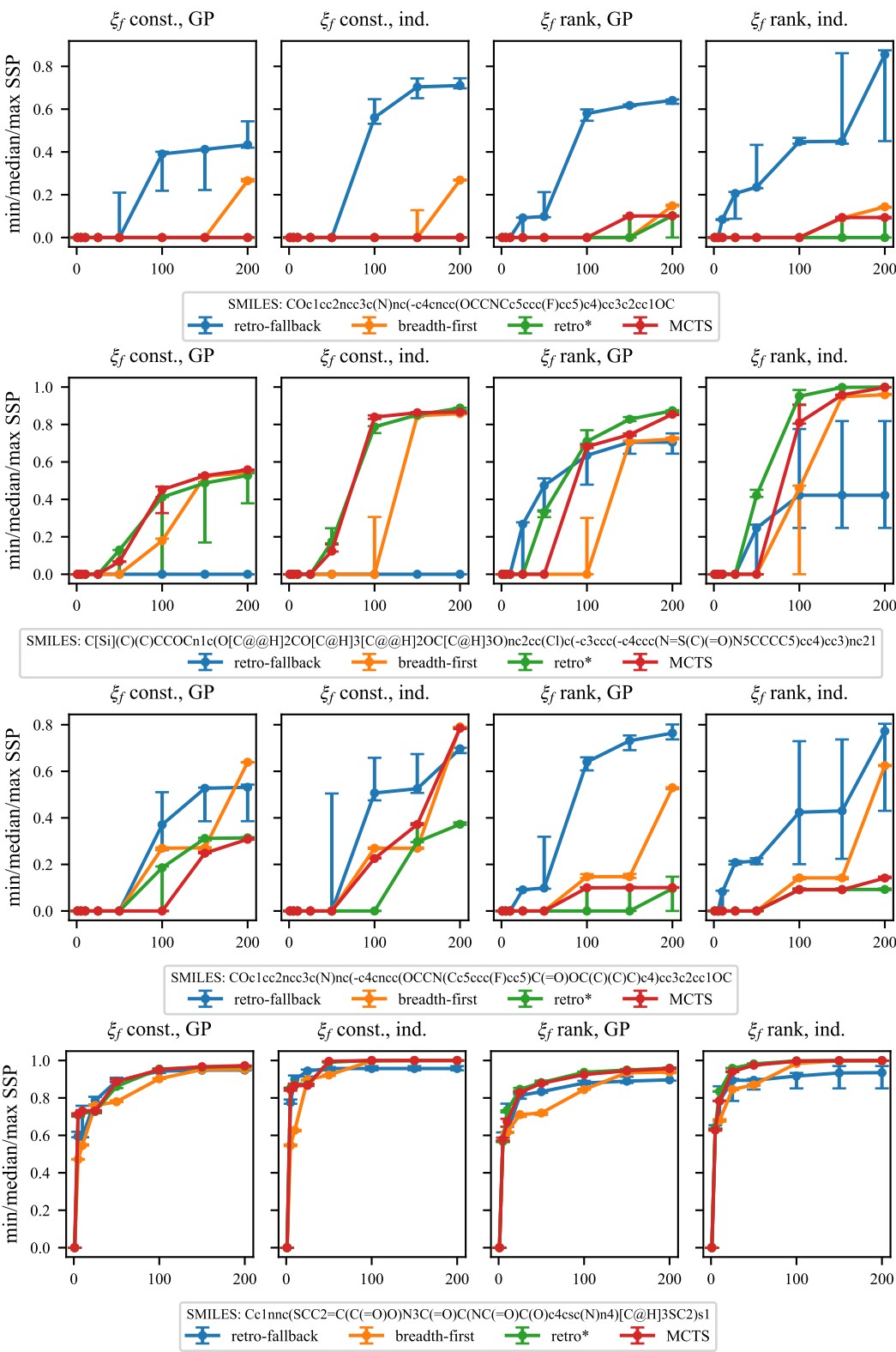

Figure G.6: Min/median/max SSP vs time for individual molecules from the 190 "hard" molecule test set (across 3 trials). Algorithms are run using the *SA score* heuristic.

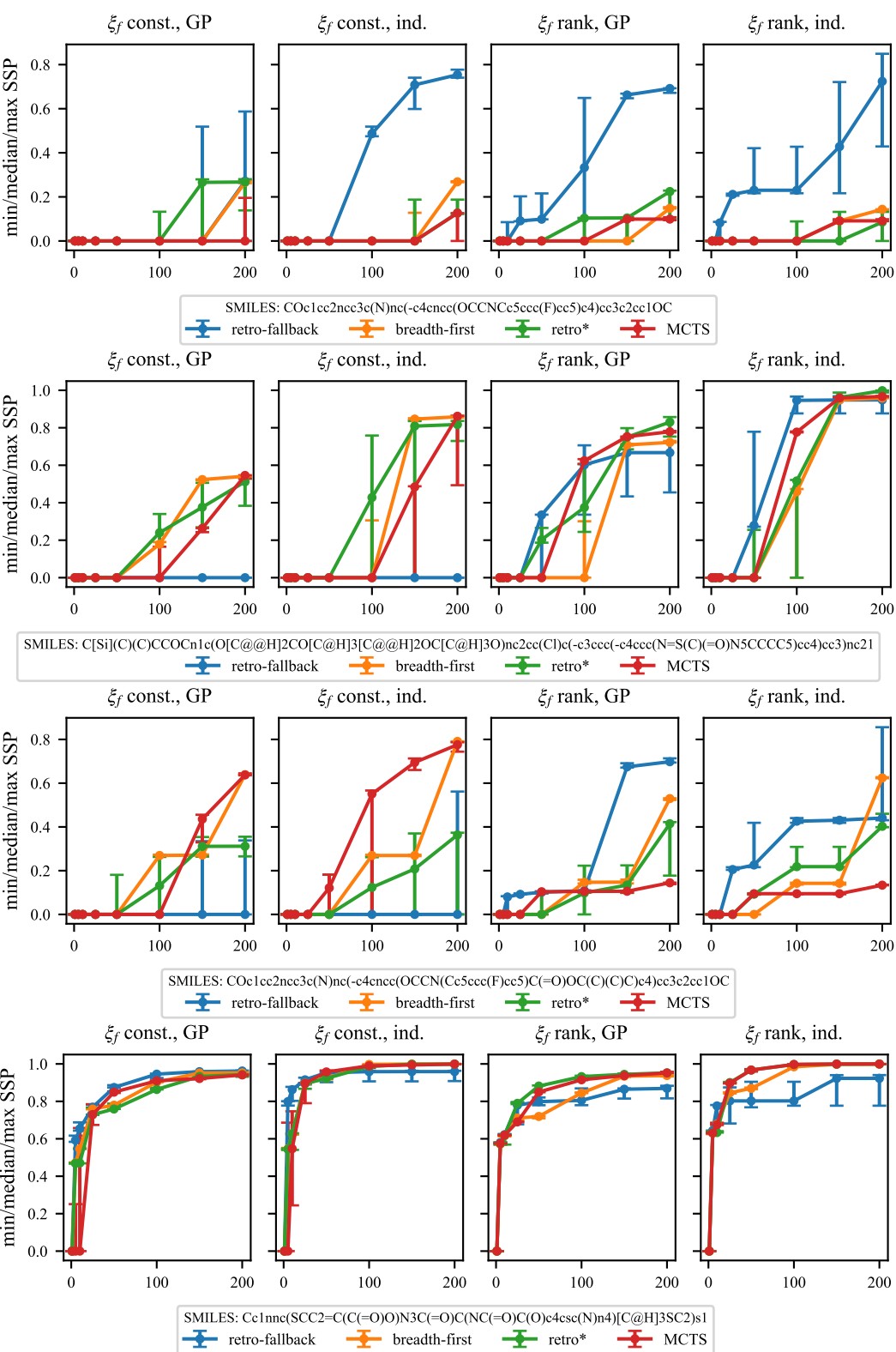

Figure G.7: Min/median/max SSP vs time for individual molecules run with the *optimistic* heuristic (same molecules as Figure G.6).

### G.2.2 RESULTS USING OTHER METRICS

Figure G.8 shows box plots of the most successful synthesis plan by the end of the search. With the exception of constant independent feasibility model on GuacaMol, retro-fallback always performs at least as well as other algorithms.

Figure G.9 shows box plots of the *shortest* synthesis plan (with non-zero success probability) found by all algorithms by the end of the search. Retro-fallback consistently performs at least as well as other algorithms.

Figure G.10 shows the fraction of molecules for which any synthesis plan with non-zero success probability is found over time. Retro-fallback consistently performs better than other algorithms.

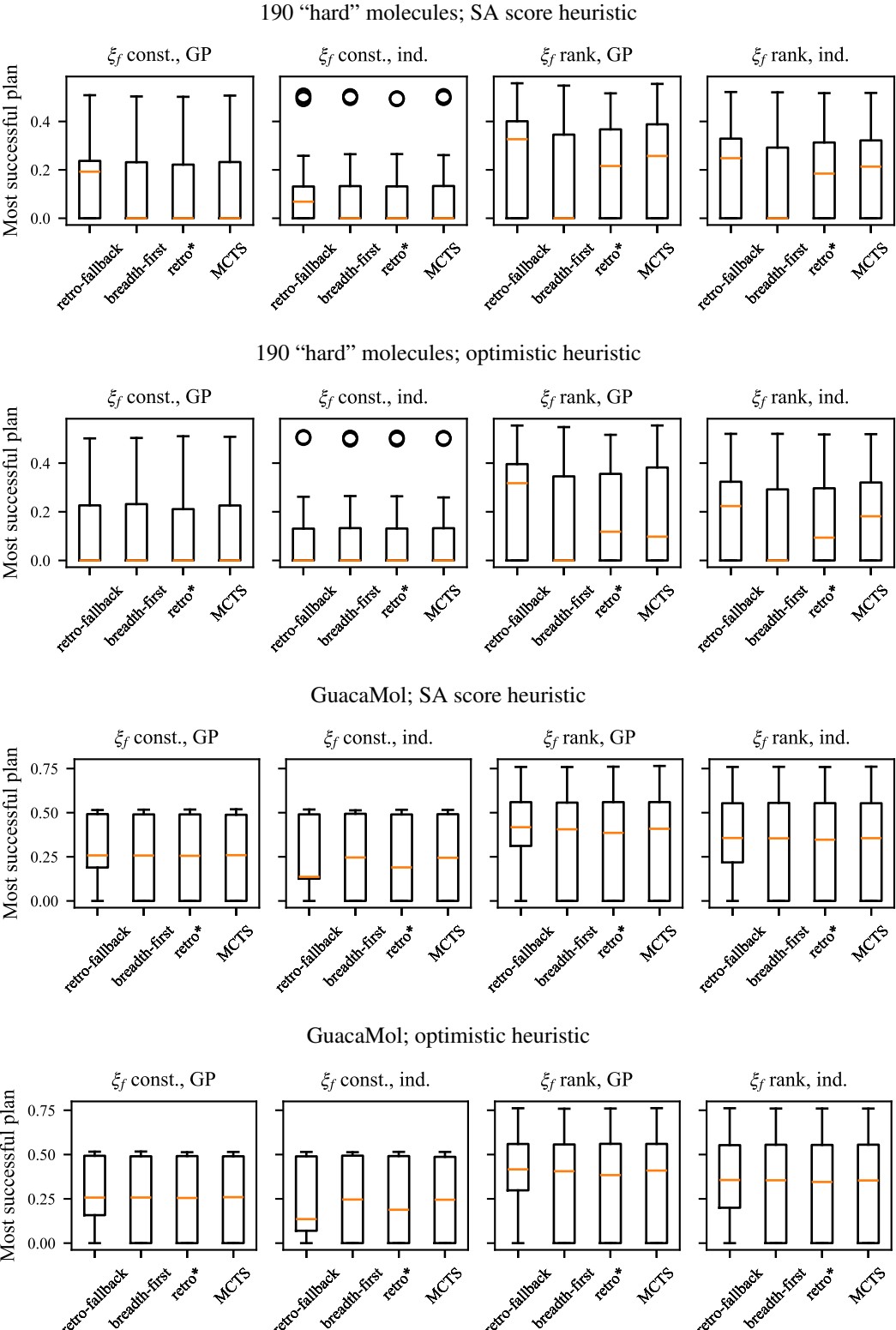

Figure G.8: Success probability of most feasible synthesis plan at end of search (i.e. $\max_{T \in \mathcal{P}_{m_\star}(\mathcal{G}')} \sigma(T; \xi_f, \xi_b)$) for different algorithms.

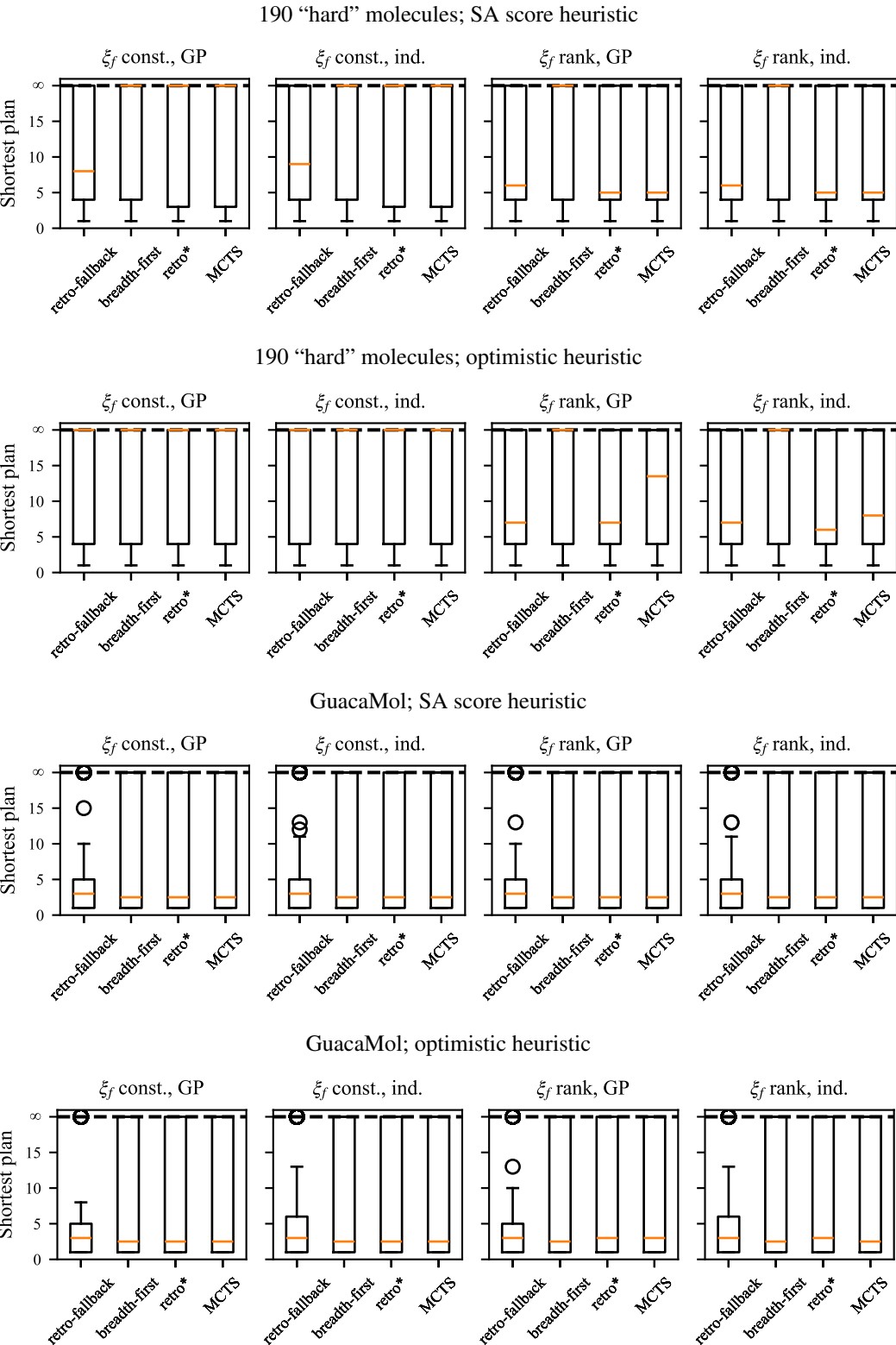

Figure G.9: Distribution of lengths of the shortest synthesis plan with non-zero success probability at end of search for different algorithms.

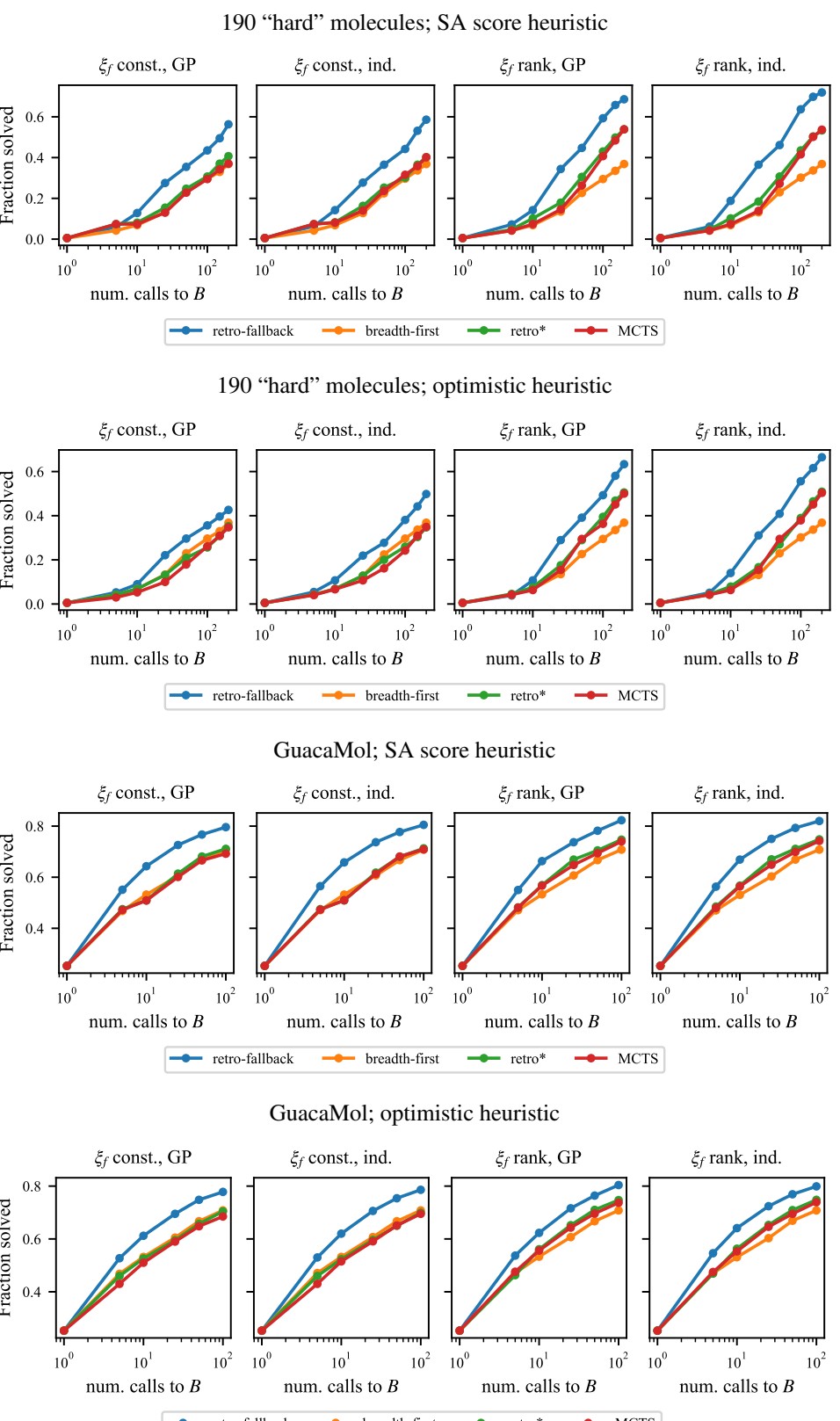

Figure G.10: Fraction of molecules for which a synthesis plan with non-zero success probability was found vs time for different algorithms.

### G.2.3 Results on FusionRetro benchmark

Here we present results on the benchmark dataset from FusionRetro, which is derived from USPTO routes and contains 5838 test molecules (Liu et al., 2023b). In addition to SSP and the fraction solved, we also evaluate performance by checking whether the outputted synthesis plans use the same starting molecules as a known ground-truth synthesis route. Liu et al. (2023b) call this metric "exact set-wise matching," but we will call it *precursor matching* because we think this name is less ambiguous. Because this metric depends on the purchasable molecules, we use a buyability model derived from the inventory of Liu et al. (2023b) instead of the model derived from eMolecules used for all other experiments. This is a deterministic model: molecules in the inventory are independently buyable with probability 1, and all other molecules are not buyable. We use the rank-independent feasibility model from section 6.2. All other details are kept the same.

The results after 50 reaction model calls are tabulated in Table G.1. As expected, retro-fallback attains higher SSP scores than the baseline retro* and breadth-first search methods, regardless of the heuristic. Just like on the GuacaMol test set from section 6.2, retro-fallback also finds at least one potential synthesis route for a higher fraction of molecules than the baselines. Finally, the precursor matching for the single most feasible synthesis plan is extremely similar for all methods (around 11%, with differences between the methods not being statistically significant).[20] This is what one would expect: the best synthesis plans found by all methods will likely be similar; the difference between retro-fallback and the other algorithms is in the secondary synthesis plans that it finds. Overall, these results show that retro-fallback outperforms baseline algorithms on the SSP metric while being no worse than the baselines on metrics which involve only single synthesis plans.

| Algorithm | Heuristic | Mean SSP (%) | Solved (%) | Precursor Match (%) |
|---|---|---|---|---|
| retro-fallback | optimistic | 67.58±0.57 | 72.03±0.59 | 10.84±0.41 |
| retro-fallback | SAScore | 68.66±0.57 | 73.19±0.58 | 10.84±0.41 |
| breadth-first | N/A | 61.66±0.61 | 65.26±0.62 | 10.96±0.41 |
| retro* | optimistic | 65.40±0.60 | 68.62±0.61 | 11.58±0.42 |
| retro* | SAScore | 65.93±0.60 | 69.10±0.60 | 11.58±0.42 |
| MCTS | optimistic | 65.02±0.60 | 68.43±0.61 | 11.20±0.41 |
| MCTS | SAScore | 65.37±0.60 | 68.64±0.61 | 11.13±0.41 |

Table G.1: Results on 5838 test molecules FusionRetro benchmark (Liu et al., 2023b). Experimental details and metrics are explained in section G.2.3. Larger values are better for all metrics. ± values indicate standard error of the mean estimate.

---

[20]Readers familiar with Liu et al. (2023b) may wonder why the precursor matching scores are much lower than what is reported in Table 1 of Liu et al. (2023b). This is because we used the same pre-trained reaction model from Chen et al. (2020) as our single-step model, whereas Liu et al. (2023b) retrains the models using their training dataset. We did not re-train the model because it also forms the basis for our feasibility model, which was loosely calibrated with the inspections of an expert chemist.

### G.2.4 RESULTS USING A NON-BINARY BUYABILITY MODEL

Figure G.11 shows the result of repeating the experiment from section 6.2 but using the "stochastic" buyability model from Appendix G.1.3. The results are essentially indistinguishable from Figure 2 and Figure G.4.

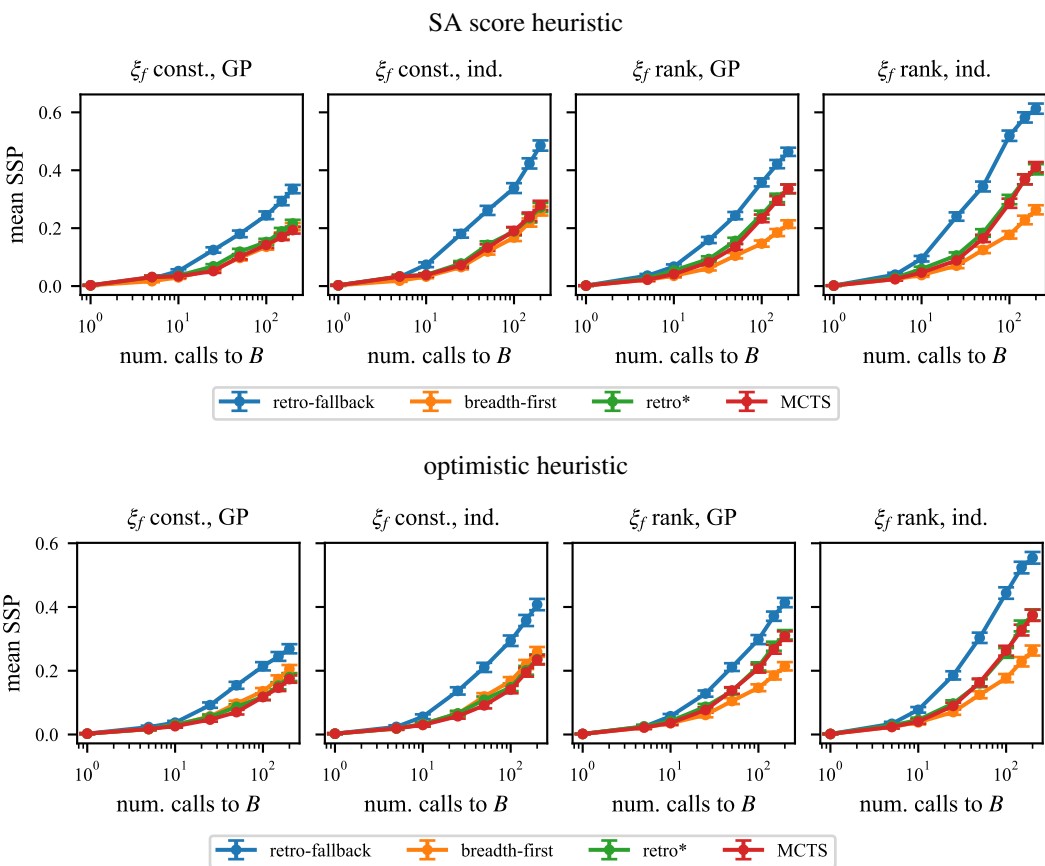

Figure G.11: SSP vs time for 190 "hard" molecules using the *stochastic* buyability model from G.1.3.

### G.3 PLOTS FOR TIME COMPLEXITY AND VARIABILITY OF RETRO-FALLBACK FOR SECTION 6.3

#### G.3.1 TIME COMPLEXITY

Figure G.12 plots the observed empirical scaling of retro-fallback with respect to the search graph size for experiments on the 190 "hard" molecule test set.

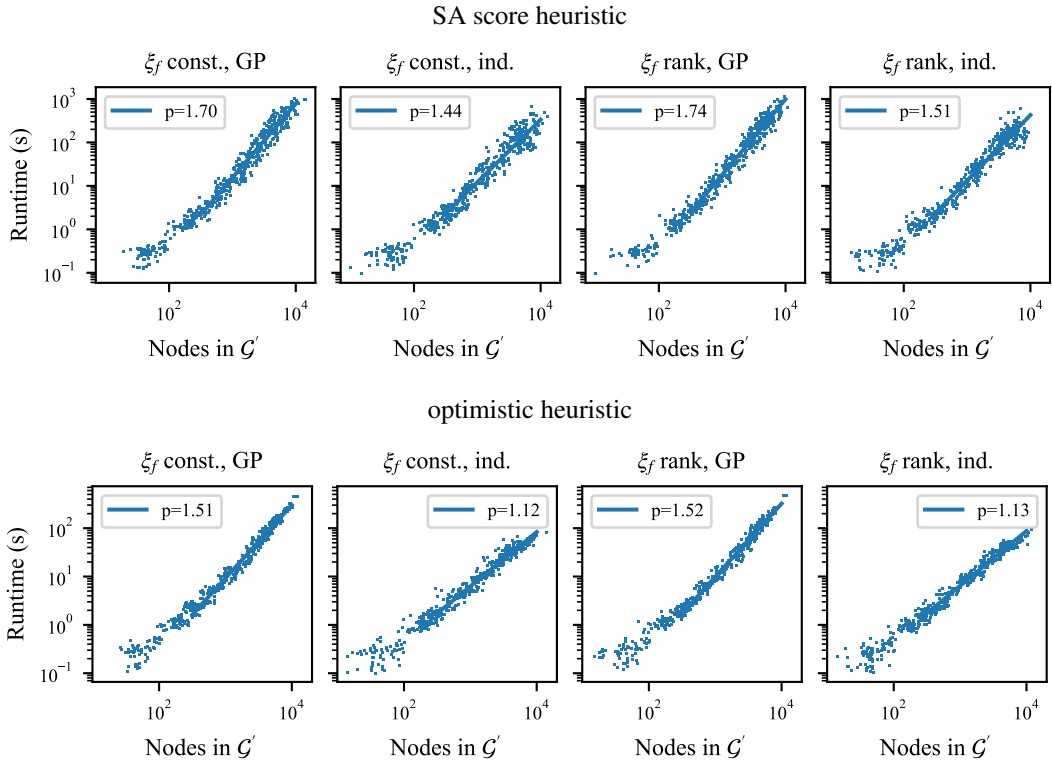

Figure G.12: Number of nodes vs total runtime for retro-fallback experiments on 190 "hard" molecule test set from section 6.2, along with log-log fit of runtime ($\log t = p \log n + C$).

#### G.3.2 VARIABILITY

Figure G.13 plots the mean and standard deviation of SSP values for a 25 molecule subset of the GuacaMol dataset. Analysis is in section 6.3.

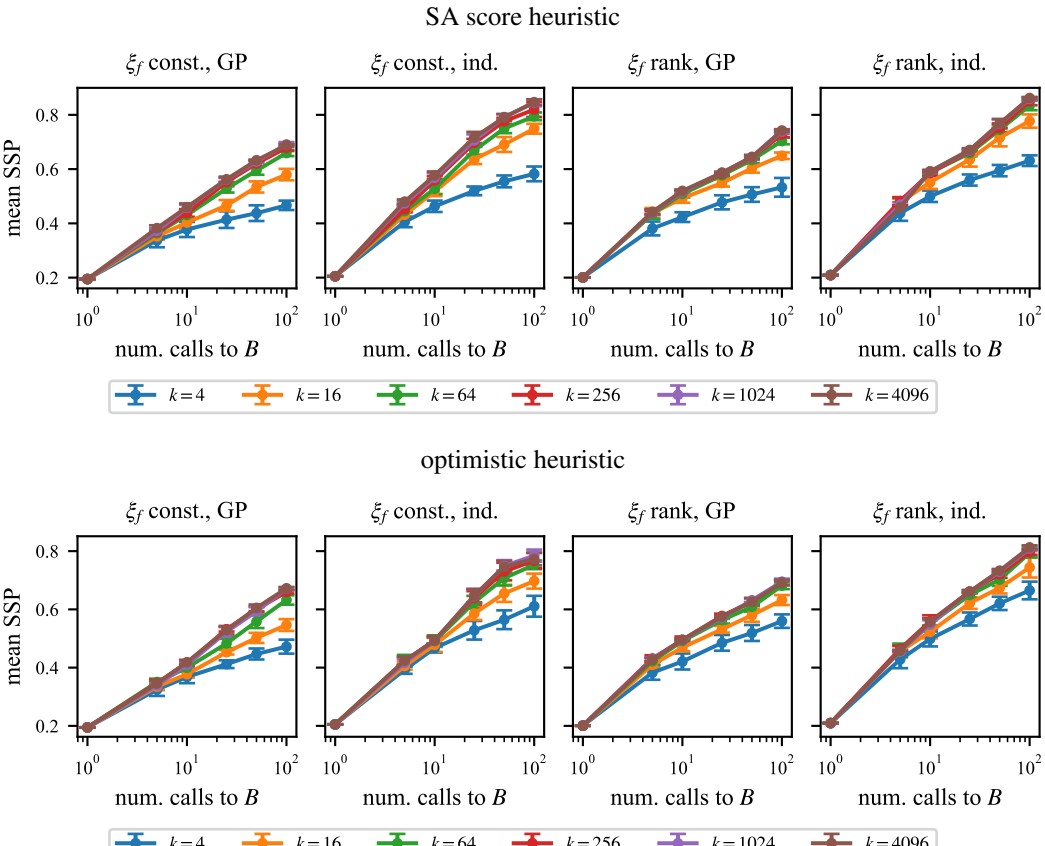

Figure G.13: Mean and standard deviation (over 10 trials) of average SSP for 25 molecules from the GuacaMol dataset when retro-fallback is run with different number of samples $k$ for the feasibility and buyability models.

# H  LIMITATIONS

Here we explicitly list and discuss some limitations of retro-fallback and SSP.

**SSP is contingent on high-quality feasibility model**   With a low-quality feasibility model, the probabilities implied by SSP will likely not be meaningful. We think the line of research advocated for in this paper will primarily be impactful *if* we are able to produce better-quality feasibility models in the future.

**Speed**   Retro-fallback appears to be significantly slower than other algorithms. This likely comes from the need to do perform updates for many samples from $\xi_f$ and $\xi_b$ (256 in most of our experiments). In contrast, other algorithms perform simple scalar updates. However, we note that if retro-fallback is used with a more complex reaction model (e.g. a transformer) then the computational cost of the search algorithm's internal computations will be less significant compared to the cost of performing an expansion. We therefore do not expect the speed of retro-fallback to limit its long-term potential utility.

**Success as a binary notion**   A lot of nuance was lost by modelling reactions and molecules as binary outcomes. In practice, chemists may care about this nuance. For example, in some circumstances having a low yield may be acceptable, but producing the wrong product may not be acceptable. Marking both outcomes as "infeasible" destroys this distinction.

**Plan length**   Chemists generally prefer synthesis plans with a few steps as possible. Retro-fallback does not directly optimize for this (and we do not see a straightforward way to extend retro-fallback to this). However, one of the main justifications for preferring short plans is that there are fewer steps that can go wrong, and therefore we expect retro-fallback to have a strong bias towards short plans regardless (similar to existing algorithms which apply a more direct penalty to the length of synthesis plans).

**Number of synthesis plans**   Our definition of SSP considers an arbitrary number of plans, in practice chemists are unlikely to try more than around 10 plans before moving on to something else. For large search graphs with many synthesis plans, SSP may therefore lose its connection with what a chemist might do in the lab. Unfortunately, we do not believe there is a straightforward way to calculate the SSP of a limited number of synthesis plans (the limit on the number of plans will likely preclude a dynamic programming solution).

## I  FUTURE WORK

**Relaxing assumptions**  If one wishes to re-insert the nuance lost by defining feasibility and buyability as binary outcomes, one could potentially explicitly model factors such as yields and shipping times and build a binary stochastic process on top of this. We do not have a clear idea of how retro-fallback or SSP could be generalized into some sort of continuous "degree of success", but imagine future work in this area could be useful. Relaxing the independence assumption of the heuristic function was discussed in Appendix C.4. The heuristic could potentially also be modified to depend on the remaining compute budget. Finally, using a separate feasibility and buyability model implicitly assumes that these outcomes are independent. We think this is a reasonable assumption because reaction feasibility is uncertain due to not fully understanding the physical system or not having a reliable model $B$, while uncertainty in buyability would originate from issues of shipping, etc. That being said, "virtual libraries" are one area where a molecule not being buyable meant that somebody else was unable to synthesize it. This may impact which reactions a chemist would predict to be feasible (although it seems unlikely in practice that a vendor would tell you the reactions that they tried). Nonetheless, if one wanted to account for this $\xi_f$ and $\xi_b$ could be merged into a joint feasibility-buyability model $\xi_{fb}$ from which functions $f$ and $b$ are simultaneously sampled.

**Theoretical guarantees of performance**  We suspect that it is possible to give a theoretical guarantee that retro-fallback's worst-case performance is better than that of retro* by formalizing the scenario in section E.3. However, we were unable to complete such a proof. We also expect it could be possible (and useful) to theoretically characterize how the behaviour retro-fallback with a finite number of samples $k$ deviates from the behaviour of "exact" retro-fallback in the limit of $k \to \infty$ where all estimates of SSP are exact. Such analysis might provide insight into how large $k$ should be set to for more general feasibility models.

**Benchmark**  If a high-quality feasibility model could be created, it would be useful to use SSP values based on this feasibility model to create a benchmark for retrosynthesis search algorithms. This might help to accelerate progress in this field, just as benchmarks have accelerated progress in other sub-fields of machine learning.

