# OpenReview forum: "Retro-fallback: retrosynthetic planning in an uncertain world"
_ICLR.cc/2024/Conference — ICLR 2024 poster_

### Official Review · Reviewer_1DzG · 2023-10-31

**Soundness:** 2 fair
**Presentation:** 3 good
**Contribution:** 2 fair
**Rating:** 6
**Confidence:** 4

**Summary:**

The submission has a clear description and clarifies that low-feasibility of retrosynthetic routes limit the real-world practicability in lab. The quantitative results and the comparison to prior methods may need to be clarified to support the statement of enhancing the generated route practicability.

**Strengths:**

1. This paper draws an insight to the unperfect reactions by the one-step prediction model and redefined the buyability of real-world molecules.

2. This paper uses a novel approach based on stochastic processes to solve the retrosynthetic problem.

**Weaknesses:**

1. More quantitative experimental results and baseline comparisons need to be clarified.

2. This paper has a solid motivation and a novel solution to perform the route planning but is limited by the proposed SSP metric. SSP attempts to calibrate the feasibility of chemical reactions. However, it will not perform better than a forward one-step prediction model since they are trained on the similar reaction dataset. Besides, SAScore is a trivial metric as it prefers unrealistic large carbon rings.

**Questions:**

The quantitative results seem not adequate enough to support the findings, could the author present more quantitative results?

**Details Of Ethics Concerns:**

NA.

---

> ### Author Response · Authors · 2023-11-18
> **Initial response to reviewer 1DzG**
>
> Thanks for reading our paper and providing helpful comments. We will respond to your comments and questions below. Please also refer to our general response ([link](https://openreview.net/forum?id=dl0u4ODCuW&noteId=GbZ5C1tSlR)).
>
> **Weakness 1: more results**
>
> We have performed _over 6000_ new search comparisons on the retro* and FusionRetro test datasets to strengthen our quantitative results. The results are shown in sections F.4-F.5 and confirm that retro-fallback is more effective at maximizing SSP than other algorithms while being no worse with respect to other metrics.
>
> We believe that these additional results are more than enough to support our claims. We remind the reviewer that previous papers in multi-step retrosynthesis (e.g. retro*, DFPN-E, PDVN) were accepted to similar conferences despite performing fewer experiments.
>
> The baselines for comparison are described in detail in sections F.1.4-F.1.5. Essentially we did our best to configure some of the most common baseline algorithms for multi-step retrosynthesis to optimize SSP (knowing it is not possible to directly set SSP to be the "objective"). We are happy to answer any specific questions you may have about them.
>
> **Weakness 2: limits of SSP metric**
>
> We think the reviewer may have misunderstood our SSP metric. First, SSP is not a single specific metric, but rather a way to translate the predictions of a reaction feasibility model into a metric for entire synthesis plans. Its values depend on an underlying reaction feasibility model, and therefore it is not an *alternative* to a forward one-step prediction model, but would likely be *implemented* using such a model.
>
> Second, SSP does not depend on SAScore in any of our experiments. We used SAScore as a _search heuristic_ to guide our algorithms, but never to evaluate the quality of the synthesis plans found during search. We agree it is trivial and has unrealistic preferences which make it unsuitable to evaluate route quality.
>
> **Overall**: we hope we have addressed your concerns, in particular your request for additional quantitative results. Please let us know if you have any follow-up questions.

---

> > ### Comment · Reviewer_1DzG · 2023-11-22
> >
> > I appreciate the authors for preparing the rebuttal, the clarifications partially addressed my concerns. After checking the rebuttal and comments from other reviewers, I would like to maintain my rating.

---

> > > ### Author Response · Authors · 2023-11-22
> > > **Thank you for considering our rebuttal. Happy to answer further questions.**
> > >
> > > Thank you for reading and considering our rebuttal. You state that our clarifications _partially_ address your concerns. May we ask which concerns are still unresolved? We are happy to answer any questions or make additional changes during the remainder of the rebuttal period to continue resolving your concerns.

---

### Official Review · Reviewer_ghCd · 2023-11-03

**Soundness:** 3 good
**Presentation:** 3 good
**Contribution:** 3 good
**Rating:** 6
**Confidence:** 3

**Summary:**

This paper aims to find multiple retrosynthesis plans to cope with the uncertainty of infeasible reactions or non-buyable molecules. To achieve this goal, the paper first presents a evaluation metric SSP to quantify the probability of at least one synthesis plan can work (judged by a feasibility and buyability model), and further designs a retrosynthetic planning algorithm retro-fallback that can greedily expand molecules to maximizes SSP, specially when all existing synthesis plans currently fail.

**Strengths:**

1. The objective for addressing the uncertainty of infeasible reactions or non-buyable molecules is well-motivated and important in practice, and the paper laid out a clear and general formalism for this objective (e.g., feasibility, buyability, SSP);
2. The path from motivation to solution is well-paved, from trivial/straightforward solution to the proposed one, and discussed the connection to related work systematically;
3. The evaluation directly answers questions about the claims made by the paper.

**Weaknesses:**

1. While I appreciate the general formulation and systematic discussion about planning algorithms, given the existence of retro*, the novelty and advantage of this work (e.g., performing parallel updates using multiple samples) is not that obvious to me. I would encourage adding more discussion when talking about detailed method.
2. Since this work focuses on multi-step retrosynthesis, related works [1] working on this should be discussed.

[1] Liu, Songtao, et al. "FusionRetro: molecule representation fusion via in-context learning for retrosynthetic planning." International Conference on Machine Learning. PMLR, 2023.

**Questions:**

1. How does Algorithm 1 generate multiple possible reaction graph for a target molecule?
2. Algorithm 1 could generate plan that is not a valid plan, e.g., when the algorithm terminates, there are still tip nodes in G'. To take such cases into consideration, can the authors also evaluate performance using two other metrics: the success rate as in Retro*, and the set-wise exact match accuracy as in FusionRetro?
3. How sensitive the method is to the number of samples when computing line 4-6 in Algorithm 1?
4. It is interesting to see retro-fallback with transformer can be more computational efficient, and the authors elaborate more on that (e.g., how to combine both for improvement)?

---

> ### Author Response · Authors · 2023-11-18
> **Initial response to reviewer ghCd**
>
> Thanks for reading our paper and providing helpful comments. We will respond to your comments and questions below. Please also refer to our general response ([link](https://openreview.net/forum?id=dl0u4ODCuW&noteId=GbZ5C1tSlR)).
>
> **Weakness 1: novelty/advantage of this work over retro\***
>
> There are two ways to view the advantage and novelty of retro-fallback over retro\*:
>
> 1. Objective: retro-fallback optimizes the SSP objective (i.e. probability that at least one synthesis route is successful) while retro* minimizes a scalar cost. Critically, _there is no way to set the costs in retro* so that it optimizes SSP_. An example which illustrates this well is when there are correlations between reaction outcomes (e.g. either reaction A or reaction B will work, but not both). This cannot be represented by assigning individual scalar costs to reactions A and B. Therefore retro-fallback _can optimize objectives which retro* is incapable of._
> 2. Mechanism: retro-fallback performs $k$ independent updates and aggregates them to estimate a conditional expectation, while retro* uses just a single cost function. Retro-fallback is a more complicated algorithm than retro\*.
>
> We have revised the comparison of retro* vs retro-fallback in section 5 to hopefully make this distinction more clear.
>
> **Weakness 2: FusionRetro not discussed**
>
> Thank you for pointing us to this work, we were not aware of it. FusionRetro was a very interesting paper that proposes a hybrid single-step model and multi-step search algorithm, as well as a new metric. We have added a citation to this work and dedicated a section in the appendix to discussing it (G.4). We have also added an experiment using their benchmark dataset and metric in apprix F.5 (described below and in our general response).
>
> **Question 1: how does algorithm 1 generate multiple reaction graphs**
>
> Much like retro*, proof number search, or MCTS, retro-fallback does not directly generate synthesis plans: instead, it grows an explicit search graph which contains the synthesis plans implicitly. A second enumeration step is required afterwards to extract the synthesis plans. Algorithm 1 just states the procedure for growing the search graph but omits this second step. We omitted it because other works (e.g. retro*) also do not discuss it and because enumerating paths in a graph can be done with relatively simple depth-first or breadth-first search algorithms.
>
> **Question 2: invalid plans/extra experiments**
>
> Our answer to Q1 should hopefully clarify this. Retro-fallback will almost certainly leave tip nodes in G, but the secondary algorithm to enumerate synthesis routes will skip over any routes whose end nodes are not purchasable. Therefore it is not possible to output invalid plans.
>
> Nonetheless we still performed an experiment with the FusionRetro dataset/metric as the reviewer suggested. Essentially we ran retro-fallback and our baseline algorithms on the FusionRetro test set using one of our existing feasibility models. The results are shown in appendix F.5 and suggest that retro-fallback achieve a higher SSP score while performing similarly on the FusionRetro exact matching metric.
>
> We further note that the “success rate” metric was already included in the paper under the name “fraction solved” (see figures F.6-F.7). Retro-fallback consistently outperformed the baseline algorithms according to this metric and did so on the FusionRetro benchmark too.
>
> **Question 3: sensitivity to number of samples**
>
> This is a good question. In figure 3 (right) we show empirically that performance generally improves as the number of samples increases, but that this effect seems to plateau after ~1000 samples. Therefore the method seems to only be sensitive when the number of samples is relatively small.
>
> **Question 4: combining with transformer**
>
> Good question: we think the best potential to combine our method with a transformer is to use it as the single-step reaction prediction model or feasibility model. By not relying on templates, such a model could be very flexible. We discuss combining retro-fallback and FusionRetro specifically in appendix G.4. This is an exciting direction for future research!

---

> ### Public Comment · ~Songtao_Liu2 · 2023-11-18
> **Thanks a lot for discussing my work**
>
> Dear Reviewer and Authors,
>
> I am Songtao Liu, the first author of FusionRetro. I noticed that you have been discussing my work, and I have carefully and thoroughly checked the results of this under-reviewed paper. I believe it is an outstanding work that will undoubtedly inspire future research in this field. I greatly appreciate your discussion of my work.
>
> Best regards,
> Songtao Liu

---

> > ### Author Response · Authors · 2023-11-20
> > **Thanks for the kind comment**
> >
> > We appreciate you reading our paper and glad that you liked it 😃

---

### Official Review · Reviewer_L6R1 · 2023-11-03

**Soundness:** 3 good
**Presentation:** 3 good
**Contribution:** 2 fair
**Rating:** 6
**Confidence:** 4

**Summary:**

This paper proposes an algorithm for retrosynthetic planning named Retro-Feedback.
The algorithm is tested on a benchmark problem and achieved improving
performance compared with existing algorithms, including Retro* and MCTS.

**Strengths:**

- The algorithm is described in detail.

- The improving performance on the benchmark problems.

- Proposed a novel measure of successful synthesis probability (SSP) for retrosynthesis.

**Weaknesses:**

- The reviewer has several questions explained in the Questions below.

**Questions:**

1. On page 4.
"... while algorithms like breadth-first search or proof-number search (Kishimoto et al., 2019) have no customizable rewards or costs of any kind."\
In Section 5.
"In fact, if costs are defined to be negative log probabilities then the updates for ψ and ρ are essentially equivalent to the “reaction number” and “retro* value” updates from (Chen et al., 2020)."\
The reviewer could not find the key difference between retro-fallback and the two existing algorithms, DFPN-E [Kishimoto+ 2019] or Retro* [Chen+ 2020].
If we use the negative log probabilities as the proof or disproof number, retro-feedback, Retro*, and DFPN-E seem very similar.\
Probably, the reviewer did not understand what this sentence meant.
"The key difference is that retro-fallback performs parallel updates..."\
What does "parallel" mean in this case?
Could the authors elaborate on this part?

1. The reviewer might have misunderstood something, but the problem described in Section   4.1 could easily avoided if the authors used a hash table for implementing the search algorithms.
(Using a hash table is a standard technique for proof-number search.)
If a hash table is used, a cycle could be easily detected, so it is possible to avoid problems caused by the same molecule or reaction appearing multiple times in a path.

1. P. 1.
"Although existing algorithms may find multiple synthesis plans, they are generally not designed to do so, and there is no reason to expect the plans found will be suitable as backup plans (e.g. they may share steps with the primary plan and thereby fail alongside it)."\
The following paper proposes an AND-OR-tree-based search algorithm for retrosynthesis, which keeps enumerating (probably) all the synthetic routes one by one in the order of some preference.
> Shibukawa, R., Ishida, S., Yoshizoe, K. et al. CompRet: a comprehensive recommendation framework for chemical synthesis planning with algorithmic enumeration. J Cheminform 12, 52 (2020). https://doi.org/10.1186/s13321-020-00452-5

Minor comments.
1. There is an older paper that proposes to use AND-OR tree search for retrosynthesis. Please consider referring to this paper.
> Heifets, A., & Jurisica, I. (2021). Construction of New Medicines via Game Proof Search. Proceedings of the AAAI Conference on Artificial Intelligence, 26(1), 1564-1570. https://doi.org/10.1609/aaai.v26i1.8331
https://ojs.aaai.org/index.php/AAAI/article/view/8331

1. Is it reasonable to assume there is always only one product?

1. P. 2. "tip nodes"
  In graph search terminology, I think this is called "frontier nodes".

---

> ### Author Response · Authors · 2023-11-18
> **Answers to questions from reviewer L6R1**
>
> Thanks for reading our paper and providing helpful comments. We will answer your questions below. Please also refer to our general response ([link](https://openreview.net/forum?id=dl0u4ODCuW&noteId=GbZ5C1tSlR)).
>
> **Q1 (key difference between retro-fallback and existing algorithms DFPN-E and Retro\*)**
>
> There are two ways to view the differences. The first is in the objective: retro-fallback aims to find a _set_ of synthesis plan which act as backup plans for each other (thereby maximizing our SSP metric) while retro*/DFPN-E aim to find a _single_ plan which minimizes some scalar cost. Importantly, this cost _cannot_ simply be set to SSP (we explain this in more detail in section 4.1 and appendix C). As an intuitive example, imagine that either reaction A or reaction B will be feasible, but not both: this dependency cannot be captured by a scalar cost. _Therefore, retro-fallback is pursuing a fundamentally different objective than the existing algorithms mentioned in your review._
>
> The second way to view the differences is mechanistically. The main difference is that retro*/DFPN-E all perform a _single_ cost update in an AND/OR graph while retro-fallback performs _$k$ separate updates_ (one for each sample from the stochastic processes). This is what we meant by “parallel updates”. Although the individual update equations are equivalent to those of retro* in one special case, they are not equivalent in general. As far as we can tell, the updates are not equivalent to DFPN-E in any cases.
>
> In summary, although these algorithms all perform heuristic-guided search on an AND/OR graph, their objectives and mechanisms are distinct, so we think the algorithms are quite different. We revised some text in section 5 to clarify this (e.g. we removed the confusing term “parallel updates”). Please let us know if this addresses your concerns.
>
> **Q2 (hash table)**
>
> We believe the reviewer may be asking about section 4.2 rather than section 4.1 because cycles are not discussed in section 4.1.
>
> If considering just a single synthesis plan, we agree that hash tables can be useful to check whether a cycle is present and potentially to remove it (in fact we use hash tables in our analysis code to enumerate valid synthesis plans). However, the problem described in section 4.2 is to calculate the recursively-defined quantities $\rho$ and $\psi$ for all nodes in the graph. We _do not_ calculate this by enumerating all plans (that would scale poorly because a graph can contain a combinatorially large number of paths). Instead we advocate for a dynamic-programming style update which will run in polynomial time. Our comment at the end of the section merely states that because the graph contains cycles, we cannot do a simple dynamic programming update in linear time. The graph contains cycles because we do not include duplicate molecule/reaction nodes like some previous works. At no point in the updating process are individual synthesis plans considered or enumerated. Therefore we don’t see how a hash table would resolve this issue.
>
> Please let us know whether this answers your question, and in particular whether you really did mean to write section 4.1.
>
> **Q3 (multiple plans)**
>
> The CompRet paper you referenced considers the problem of _extracting_ individual synthesis plans from a search graph, while retro-fallback (and retro*, MCTS, DFPN-E) are all algorithms to _construct_ a search graph. Algorithms like CompRet are required to extract the routes afterwards. We used a simple best-first search algorithm to do this but will look into CompRet as an alternative (and have added a citation to this paper). Thanks for the reference!
>
> **Minor comments**
>
> - Older paper: thanks for the reference, we have added a citation in section 2. We were not aware of this paper.
> - Assuming one product: all existing multi-step retrosynthesis algorithms (e.g. retro*, DFPN-E) make this assumption, even if they do not state it explicitly. We think it is reasonable since many reactions have a single main product, and multi-product reactions can be represented as two separate reactions (e.g. A+B -> C+D can become A+B->C and A+B->D).
> - Tip nodes: we copied the usage of this term from Jiménez and Torras (2000) but are happy to use the term “frontier node” if you think that is clearer.

---

> > ### Comment · Reviewer_L6R1 · 2023-11-22
> > **Reply to the answers.**
> >
> > Thank you for the detailed feedback.
> > I still have some questions.
> >
> > 1. Does this phrase, "either reaction A or reaction B will be feasible, but not both," mean that using some of the pairs of chemical reactions is not feasible? Sorry for my lack of chemistry knowledge, but I was unaware of such cases.
> > I think your answer resolves Q1.
> >
> > 1. I was asking about section 4.2. Sorry for confusing you. I am not sure if I understood the answer. To my knowledge, from the algorithm viewpoint, existing algorithms such as df-pn, df-pn+, DFPN-E, and CompRet detect duplicate nodes correctly by using a hash table. I apologize for asking on the last day, but I could not understand the sentence, "The graph contains cycles because we do not include duplicate molecule/reaction nodes like some previous works." What do the authors mean by "some previous works"?
> >
> > 1. If I understood correctly, CompRet constructs and extracts the synthesis plans. Please clarify if this is wrong.

---

> ### Author Response · Authors · 2023-11-22
> **Polite follow up on our rebuttal**
>
> We thank again reviewer L6R1 for their effort during the reviewing process. We believe we have addressed the stated concerns with our response ([link here](https://openreview.net/forum?id=dl0u4ODCuW&noteId=ToJgkKyh7Y)) and we would like to ask the reviewer if they think this is the case as well. If the response is affirmative, and based on the higher ratings provided by the other reviewers, we would like to ask the reviewer if they are happy to increase their score.

---

> ### Author Response · Authors · 2023-11-22
> **Further answers for reviewer L6R1**
>
> Thank you very much for responding to our rebuttal. We saw that you increased your score and appreciate that. We will try to answer your extra questions.
>
> **Q1: different reactions being feasible**
>
> This is a fair question and we understand your confusion. We will try to give a much more detailed answer.
>
> The case we were referring to was a case where a feasibility model predicts that either a reaction A is feasible or a reaction B is feasible, but likely not both. For example, imagine a chemist knows that the reaction $P+Q$ will produce either $R$ (reaction A) or $S$ (reaction B) but doesn't know which one. An example probability distribution over feasibility outcomes which quantifies this is:
>
> |Outcome|Probability|
> |-----|-----|
> |A,B both feasible|0.0|
> |A feasible, B infeasible|0.5|
> |A infeasible, B feasible|0.5|
> |A, B both infeasible|0.0|
>
> (Note that the probabilities above sum up to 1)
>
> So yes, using some of these pairs would not be feasible together.
>
> **Q2: de-duplication**
>
> We think there might be confusion between two kinds of cycles. The first kind of cycle is if a route contains two reactions which undo each other, e.g. $A\rightarrow B \rightarrow A \rightarrow C$. The second kind of cycle is if there exists a path in the serach graph $\mathcal{G}'$ which revisits a node.
>
> The previous works we refer to (specifically retro* and DFPN-E) use represent the search graph as a _tree_, which by construction will never contain the second kind of cycle. They construct a tree by allowing duplicate nodes in the graph. For example, if the reactions $B+C\rightarrow A$ and $D+C\rightarrow A$ are both in the search graph, there will be two $C$ nodes added to the graph. To avoid synthesis routes which contain the first kind of cycle, hash tables can be used to track which reactions have already been visited so that nodes which would imply a cyclic route are not added.
>
> In our implementation of retro-fallback we use a search graph which only contains one node for each molecule, and is therefore not a tree. Figure B.1 shows an example of such a graph (which does not contain a cycle). However, in general it is possible for such graphs to contain a cycle if reactions like $A\rightarrow B$ and $B\rightarrow A$ are both possible. Our discussion of cycles pertains to _this_ type of cycles. Identifying whether cycles are present is still easy (and can be done with the hash table), but simply identifying the cycles does not yield a way to solve the recursive update equations for $\psi$ and $\rho$. Does that make sense?
>
> **Q3: CompRet**
>
> Sorry, we made a small mistake. Quoting the intro of this paper:
>
> > CompRet implements the following three steps to recommend synthetic routes: (1) constructing a chemical reaction network based on the depth-first proof number search (DFPN) and template-based retrosynthesis [29, 30], without a large chemical reaction network such as the NOC, (2) enumerating all synthetic routes from the network using a novel algorithm, and (3) recommending multiple synthetic routes by developing a naive visualization method and simple score functions.
>
> When we described CompRet in our revised paper, we were only referring to step (2) because that was the novel algorithm proposed in the paper. However, looking again it seems that CompRet refers to the combination of all 3 steps. Apologies for the confusion. Still, DFPN and the route extraction regime in CompRet only consider metrics for individual routes and do not appear to have any mechanisms to return secondary routes which are suitable _backup plans_ for the first route, which is retro-fallback's goal. Does this match the reviewer's understanding of CompRet?

---

> > ### Comment · Reviewer_L6R1 · 2023-11-23
> > **Reply to the further answers**
> >
> > I thank the authors for the quick reply.
> > Q1 and Q3 are resolved.
> >
> > Q2,
> > I was not aware that DFPN-E allows the second type of cycles. Thank you for the explanation. But at least, the original df-pn variants do not allow duplicate nodes. Otherwise it would be a problem when solving two player games. Do I misunderstand something specific to chemistry?

---

> > > ### Author Response · Authors · 2023-11-23
> > > **Response to Q2 from reviewer questions**
> > >
> > > In your previous response you wrote:
> > >
> > > > Q2, I was not aware that DFPN-E allows the second type of cycles.
> > >
> > > I think you may have misunderstood us: as far as we understand DFPN-E uses a _tree_ and therefore _does not_ contain the second kind of cycles.
> > >
> > > We are unsure if other df-pn variants allow duplicate nodes or not. We are not super familiar with the literature on proof number search. It could be the case that the presence of absence of cycles does not affect the calculation of proof numbers very much so that using a graph containing cycles does not cause problems (it causes problems in retro* and retro-fallback because the values depend on finding acyclic paths in the graph). It is also possible that graphs of mathematical theorems simply do not contain many cycles (i.e. there are not many pairs of theorems $A \Rightarrow B,\ B\Rightarrow A$). However, in chemistry reaction pairs $A \rightarrow B,\ B\rightarrow A$ are very common so the graph naturally has a lot of cycles.
> > >
> > > If you think it would be appropriate we can add some of this context to the related work section in Appendix C where we comment on proof number search in more detail?

---

### Author Response · Authors · 2023-11-18
**Initial response to all reviewers**

Thank you all for your helpful comments and feedback. We have just posted a revised version of our paper and will shortly post an individual reply to each reviewer answering their individual questions. Here we wish to summarize the reviews, our responses and the changes made to our paper.

**Compliments:** We appreciated many positive comments left by the reviewers, including a novel and well-motivated objective, clear explanation of our algorithm, and positive experimental results which directly support the claims in our paper. We appreciate the largely positive tone of the reviews!

**Revision:** based on reviewer comments we have made the following changes to our paper pdf.

1. Revised the discussion comparing retro-fallback to existing algorithms in section 5 to better explain the novelty and advantage of our approach.
2. Performed new experiments on _over 6000 test molecules_ to strengthen our quantitative results:
   - In appendix F.4 we present results for the 190 “hard” molecules from Chen et al 2020. Retro-fallback significantly outperforms the other baselines for all feasibility models.
   - In Appendix F.5 we present results on 5838 test molecules from the FusionRetro benchmark, evaluating with SSP (our metric), fraction solved, and the precursor-matching metric proposed in the FusionRetro paper. Retro-fallback outperforms baselines on SSP and performs at least as well as baselines on all other metrics.
3. Added some citations to relevant papers pointed out by reviewers, including an extended discussion of FusionRetro in Appendix G.4.

These changes are highlighted in red to be clearly visible. We believe the revised paper is much stronger and convincingly addresses all reviewer concerns. Please do let us know if there are any remaining concerns and we will try to address them within the rebuttal period. If not, then we kindly request the reviewers to reconsider their scores.

---

### Meta-Review · Area_Chair_414n · 2023-12-11

**Metareview:**

In this paper, the authors first propose a new metric to evaluate the quality of the proposed synthesis path. However, optimizing such a metric is computational intractable. Therefore, the authors proposed a heuristic searching algorithm, inspired by existing retro*, and conducted experiments to demonstrate the advantages.

The paper is interesting and empirically promising.

**Justification For Why Not Higher Score:**

There are several issues pointed out by the reviewers. The major concern is the discussion and comparison with the existing methods, e.g., retro*, DFPN-E, and FusionRetro, should be emphasized in main text to further position the method well in the literature.

Although this has been provided in the revision, the novelty of the proposed method may not enough for a spotlight.

**Justification For Why Not Lower Score:**

The proposed method is interesting and promising in practice.

---

### Decision · Program_Chairs · 2024-01-16

Accept (poster)